# CETN3 deficiency induces microcephaly by disrupting neural stem/progenitor cell fate through impaired centrosome assembly and RNA splicing

Jing Xu [1,16], Xiao Mao[2,16], Zhen Liu [2,16], Na Jiang[1], Xin E Wong[1], Deng Liu [1,3], Yuan Wang[1], Huaizhe Zhan[1], Shiyi Liu[4], Jiayao Yu[1], Ruiying Yuan[5,6], Qingran Bai [7], Xianshu Bai [8,9,10], Wenhui Huang[8,9,11], Ruoxiao Xie [12], Veronica Krenn [13], Frank Kirchhoff [8,9], Hua Wang[14✉], Zhenming Guo [1✉] & Shan Bian [1,2,15✉]

## Abstract

Primary microcephaly, a rare congenital condition characterized by reduced brain size, occurs due to impaired neurogenesis during brain development. Through whole-exome sequencing, we identified compound heterozygous loss-of-function mutations in *CENTRIN 3* (*CETN3*) in a 5-year-old patient with primary microcephaly. As *CETN3* has not been previously linked to microcephaly, we investigated its potential function in neurodevelopment in human pluripotent stem cell-derived cerebral organoids. We showed that *CETN3*-knockout (KO) organoids successfully recapitulated the microcephaly phenotype of reduced size compared to the control organoids. Through transcriptomic, histological, and protein analyses, we found that *CETN3* deficiency directly interferes with neuronal differentiation and reduces proliferative capacity in neural stem/progenitor cells by impairing centrosome assembly required in cell cycle progression, consequently activating apoptosis. Furthermore, our data uncovered previously undocumented indirect effects of CETN3 through interaction with RNA splicing machinery involved in brain development. These findings expand the scope of known regulatory mechanisms of CETN3 in brain development and its etiological roles in human brain malformation.

**Keywords** CETN3; Microcephaly; Neurogenesis; Centrosome Duplication; RNA Splicing
**Subject Category** Neuroscience

## Introduction

Microcephaly is a relatively rare defect defined by occipital-frontal head circumference at least two standard deviations (SD) smaller than the average (Becerra-Solano et al, 2021), occurring in approximately one case per several thousand births, and manifesting as either an isolated condition or accompanied with other clinical features such as intellectual disability, short stature, ataxia, and seizures (Karaer et al, 2022; Mumtaz et al, 2015). Known causes include prenatal infections (e.g., toxoplasmosis or Zika virus), exposure to toxic chemicals, genetic abnormalities, and severe malnutrition. Based on the timing of onset, primary microcephaly is a congenital condition, whereas secondary microcephaly occurs postnatally, with affected individuals failing to exhibit normal growth in head size during early childhood (Woods, 2004). In general, the majority of primary microcephaly cases have been attributed to genetic abnormalities, with 32 genes known to share a causal association with microcephaly primary hereditary (MCPH) according to the OMIM database (http://omim.org/) and published work (Asif et al, 2023; Farcy et al, 2023; Lim, 2023). Most genes linked to MCPH are involved in regulating cell cycle, centrosome assembly or function, DNA repair, and apoptosis (Naveed et al, 2018). The reduced size of primary microcephalic brains has been identified as an effect of decreased neuron

[1]Institute for Regenerative Medicine, Medical Innovation Center and State Key Laboratory of Cardiovascular Diseases, Shanghai East Hospital, National Stem Cell Translational Resource Center & Ministry of Education Stem Cell Resource Center, Frontier Science Center for Stem Cell Research, School of Life Sciences and Technology, Tongji University, Shanghai, China. [2]National Health Commission Key Laboratory of Birth Defect Research and Prevention, Hunan Provincial Maternal and Child Health Care Hospital, University of South China, Changsha, China. [3]School of Basic Medical Sciences, Harbin Medical University, Harbin, China. [4]Department of Biology, Brandeis University, Waltham, MA, USA. [5]Department of Medicament, College of Medicine, Tibet University, Lhasa, China. [6]Research Center for Ecological Resources and Modern Development of Endemic Medicinal Plants on the Qinghai-Tibet Plateau, Tibet University, Lhasa, China. [7]Key Laboratory of Spine and Spinal Cord Injury Repair and Regeneration of Ministry of Education, Orthopaedic Department of Tongji Hospital, School of Medicine, Tongji University, Shanghai, China. [8]Molecular Physiology, Center for Integrative Physiology and Molecular Medicine (CIPMM), University of Saarland, Homburg, Germany. [9]Center for Gender-specific Biology and Medicine (CGBM), University of Saarland, Homburg, Germany. [10]State Key Laboratory of Natural Medicines, Department of Pharmacology, School of Pharmacy, China Pharmaceutical University, Nanjing, China. [11]Brain Research Center, Sun Yat-Sen Memorial Hospital, Sun Yat-Sen University, Guangzhou, China. [12]Department of Materials, Design and Manufacturing Engineering, School of Engineering, University of Liverpool, Liverpool, UK. [13]Department of Biotechnology and Biosciences, University of Milano-Bicocca, Milano 20126, Italy. [14]Clinical Medical Research Center for Hereditary Birth Defects and Rare Diseases in Hunan Province, The Affiliated Children's Hospital of Xiangya School of Medicine, Central South University, Changsha, Hunan, China. [15]China Regional Research Center, International Center for Genetic Engineering and Biotechnology, Taizhou, China. [16]These authors contributed equally: Jing Xu, Xiao Mao, Zhen Liu. ✉E-mail: FX20240002@csu.edu.cn; spring_gzm@tongji.edu.cn; shan_bian@tongji.edu.cn

numbers, resulting from defects in neural stem/progenitor cell (NS/PC) proliferation and differentiation, or excessive activation of apoptosis in neural progenitors during early neurogenesis (Kadir et al, 2016).

First isolated from the chlorophycean green algae, *Tetraselmis striata*, centrins belong to the EF-hand superfamily of calcium-binding proteins, and contain four calcium-binding EF-hand domains. Mice harbor four centrin-encoding genes (*Cetn1-4*), while human beings have three (*CETN1-3*) (Martinez-Sanz and Assairi, 2016; Moretti et al, 2023; Salisbury et al, 1984). The expression patterns of *CETN* vary depending on tissue type and developmental stage: *CETN1* is highly expressed in male germ cells, while *CETN2* and *CETN3* are ubiquitously expressed in somatic cells (Dantas et al, 2011; Ying et al, 2019). CETN3 localizes to both the distal end and the central core of the centriole, where it plays a role in ciliogenesis (Ying et al, 2019). Additionally, CETN3 has been reported to serve as a component of the transcription and export complex 2 (TREX-2) in the nucleus, participating in mRNA export (Schubert and Köhler, 2016; Umlauf et al, 2013). CETN3 is considered critical for centrosome duplication in lower eukaryotes, such as *Saccharomyces cerevisiae* (Schiebel and Bornens, 1995). However, its role in centrosome duplication in animals remains controversial (Dantas et al, 2012; Middendorp et al, 2000; Salisbury, 2007). Several studies have explored the CETN3 function in animal cells, but varied in their conclusions. Dantas et al demonstrated that *Cetn3* deletion in DT40 cells did not affect cell cycle, centrosome reduplication, or centrosome ultrastructure (Dantas et al, 2011). Similarly, Middendorp et al reported that *CETN3* overexpression induced no obvious mitotic defects in HeLa cells, but its overexpression in two-cell stage *Xenopus laevis* embryos resulted in insufficient cleavage and inhibited centrosome duplication (Middendorp et al, 2000). In contrast, Sawant and colleagues found that *CETN3* depletion in HeLa cells promoted centriole reduplication, and *CETN3* overexpression in U2OS cells blocked centrosome reduplication (Sawant et al, 2015). These conflicting results suggest that CETN3 might exhibit context-dependent functions in centrosomes, which could differ among species and cell types. However, no studies to date have explored the role of CETN3 in neurodevelopment, especially neocortical development.

In this study, we identified biallelic mutations in *CETN3* in a patient diagnosed with primary microcephaly. Using human cerebral organoids (hCOs) derived from human embryonic stem cells (hESCs) or induced pluripotent stem cells (iPSCs), we found that *CETN3* knockout or loss-of-function mutants could recapitulate the microcephaly phenotype, which led us to investigate its function in neurodevelopment. Using this model, our findings show that CETN3 regulates brain size, directly by regulating centrosome formation required for NS/PC proliferation, differentiation, and apoptosis, and also through a novel indirect mechanism involving interaction with RNA splicing machinery responsible for processing mRNAs of centriole-associated genes.

# Results

## Identification of pathogenic compound heterozygous *CETN3* mutations in a microcephaly patient

A microcephaly case was first clinically diagnosed by prenatal ultrasound in our clinic, based on presentation with significantly reduced head circumference (2 SD < the average; Appendix Fig. S1A) at the gestational age of 30 weeks. At birth, her head circumference measured 31.5 cm (−2 SD), but her length (48.0 cm) and weight (3.1 kg) were within the normal range for age, gender and population averages (Fig. 1A,B; Appendix Fig. S1B). At one month, her head circumference remained small (34.0 cm; −2 SD), and the patient displayed diminished pupillary light reflex and upper limb tremors induced by light exposure. Head MRI revealed bilateral lateral ventricular enlargement, with no other significant abnormalities detected (Appendix Fig. S1C). Her motor development was significantly delayed; she achieved independent sitting at 14 months and walking at 20 months, albeit with marked instability. Currently, despite normal vision and hearing, she exhibits pronounced horizontal nystagmus accompanied by head swaying, but without limb tremors. Growth records indicate that her height and weight were consistent with age-matched peers for most of her developmental trajectory. At 5-years-old, the patient's head circumference was 3 SD < the average (Fig. 1A,B; Appendix Fig. S1B), but still without other significant clinical symptoms detected, indicating it was a case of isolated primary microcephaly.

As prenatal genetic screening detected no potential pathogenic variants in known causal primary microcephaly genes, we conducted whole-exome sequencing (WES) in the patient and her parents. The WES revealed biallelic mutations in *CETN3* (GenBank: NM_004365.2), comprising a 2 bp deletion in exon 2 (c.43_44del; p.Lys15Glufs*9) of one allele and a 4 bp deletion in exon 2 (c.118_121del; p.Asp40Lysfs*5) of the other allele. Both of the *CETN3* mutations induce a frameshift leading to a premature STOP codon, which were detected in the patient's father and mother, respectively (Fig. 1C,D). Western blot confirmed that CETN3 was absent in the patient (Fig. 1E). Protein sequence alignments revealed that CETN3 is highly conserved among animals, especially in mammals (Fig. 1F; Appendix Fig. S1D,E), supporting its essential function in development. Publicly available scRNA-seq data from fetal human brain and developing mouse cortex (Li et al, 2020; Nowakowski et al, 2017) indicated that *CETN3* is ubiquitously expressed across all cell types in both species (Fig. 1G,H). Immunofluorescence (IF) staining of CETN3 in both human (gestational week 10, GW10) and mouse (embryonic day 13.5, E13.5) brains confirmed its broad expression, with the strongest signal observed along the apical surface of the ventricular zone (VZ) (Fig. 1I). Further IF co-staining assays in hCOs (day 45) and mouse cortex (E13.5) revealed CETN3 colocalized with centrosome marker γ-tubulin (Fig. 1J,K), which was consistent with previous studies that showed CETN3 localized to the centriole and transition zone of primary cilia (Ying et al, 2019).

## CETN3 ablation leads to reduced size in human cerebral organoids

To determine whether CETN3 deficiency was responsible for the reduced head circumference of our patient and to investigate its possible role in brain development, we generated *CETN3* knock-out (KO) cell lines based on H9 hESCs using the CRISPR-Cas9 system with two sgRNAs respectively targeting *CETN3* intron 1–2 and intron 2–3 to excise exon2 (Appendix Fig. S2A). The exogenous DNA fragment introduced for clone selection was subsequently eliminated using Cre recombinase. (Fig. 2A). These modifications caused premature termination of CETN3 translation. The resulting

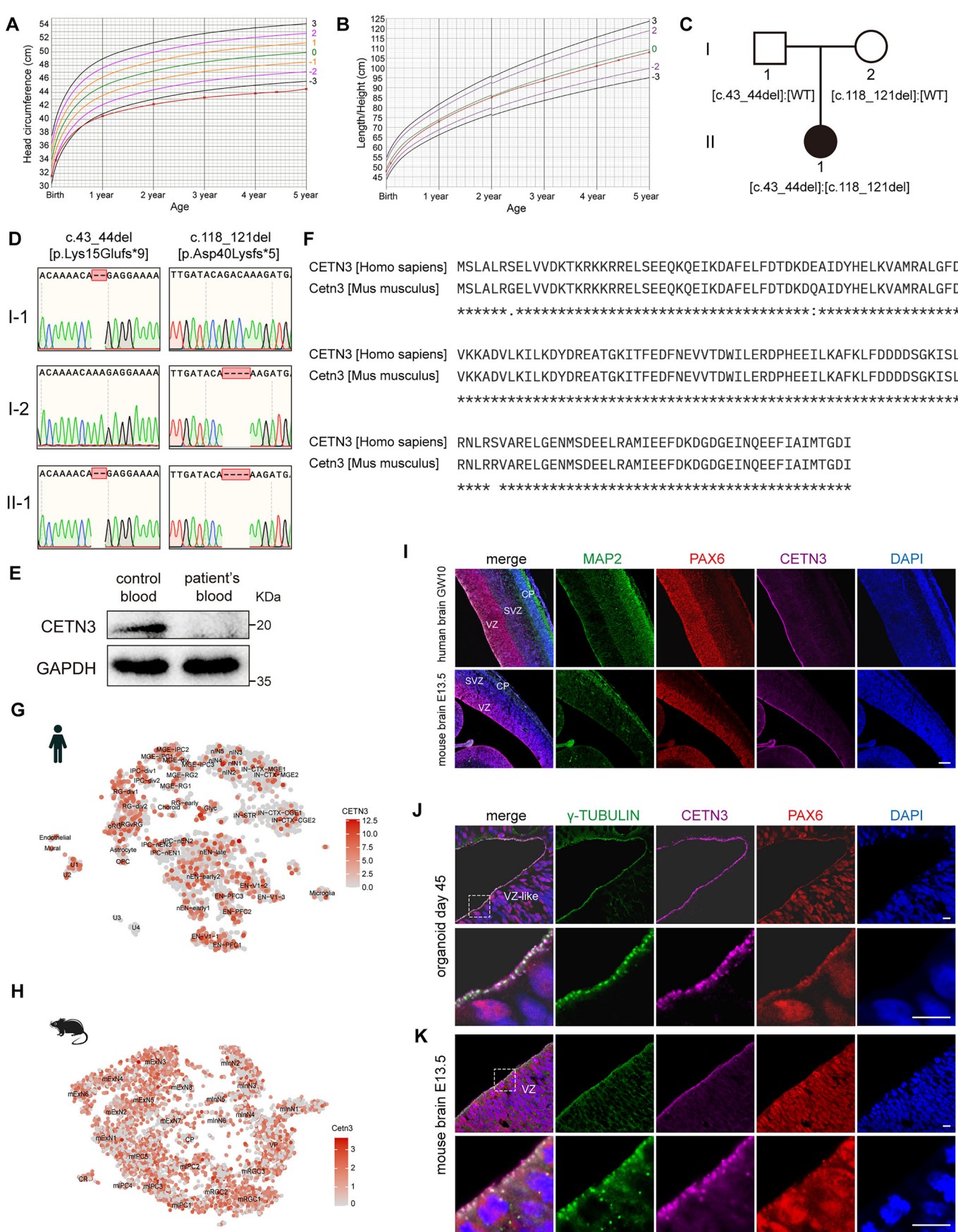

◄ **Figure 1. A patient carrying a *CETN3*-loss mutations presents with microcephaly.**

(A, B) Head circumference and height of the patient. Red lines with forks represent data of the patient, and the numbers on other lines indicate SD from the average. (C) Pedigrees of the patient's family. Squares represent male family members, while circles represent female members. The solid symbol denotes affected individuals. *CETN3* mutations carried by family members are indicated beneath each individual. (D) *CETN3* mutations carried by the family members were confirmed by Sanger sequencing. (E) Western blot verified that the CETN3 protein was lost in the patient's blood. The binding site of the antibody is located in the central region of CETN3. (F) CETN3 protein sequence alignment of human and mouse using MUSCLE 3.8. (G, H) T-SNE of public single-cell RNA-seq data from UCSC illustrates the expression of *CETN3* in human fetal cortex (G http://cells.ucsc.edu/?ds=cortex-dev&gene=HES1&cell=S46.C8) and developing mouse cortex (H http://cells.ucsc.edu/?ds=mouse-dev-neocortex). (I) Immunofluorescence of coronal cryosection from GW10 human brain and E13.5 mouse brain. Markers for neurons (MAP2) and NS/PCs (PAX6), and CETN3 were co-stained. VZ ventricular zone, SVZ subventricular zone, CP cortical plate. Scale bar: 100 μm. (J, K) Immunofluorescence of cryosection from day 45 cerebral organoids (J) and E13.5 mouse brain (K). The dashed areas in the upper panels are shown in higher magnification in the lower panels. Centrosome marker γ-tubulin was co-located with CETN3. Scale bar: 10 μm. Source data are available online for this figure.

homozygous *CETN3*-KO hESC lines, #7-5 and #12-3, were subsequently confirmed through genotyping by PCR and Western blot (Appendix Fig. S2B; Fig. 2B).

To assess the impact of CETN3 deletion on hESCs, we first conducted qPCR for the pluripotency genes (*NANOG*, *OCT4*, *SOX2*, *DNMT3B*, *TERT*, and *REX1*) (Appendix Table S1), and IF staining for the pluripotency markers (SOX2, OCT4, and NANOG) to evaluate the pluripotency of hESCs, which showed no obvious impairment (Fig. 2C; Appendix Fig. S2C,D). Subsequent IF analysis of cell proliferation and apoptosis markers, Ki67, phosphorylated histone H3 (PH3), and cleaved caspase-3 (Fig. EV1A,B), as well as EdU labeling to monitor cell proliferation (Fig. EV1B), showed that *CETN3* deletion did not affect the proliferative capacity of hESCs nor induce hESC apoptosis (Fig. EV1A–D). However, in *CETN3*-KO cells, we observed a slight increase in aberrant spindles with more than two poles when staining for microtubules and centrosomes with antibodies targeting α-tubulin and PCNT, respectively (Fig. EV1E), although such spindles accounted for a very low proportion (1–2%) (Fig. EV1F).

To further investigate the possible function of CETN3 in microcephaly, we generated hCOs, a well-established model for studying human brain development and developmental disorders, from both wild-type (WT) H9 and *CETN3*-KO hESCs (Fig. 2D) (Farcy et al, 2022; Lancaster et al, 2013; Wang et al, 2023). Monitoring of hCO morphology and size throughout the culture period (Fig. 2E) showed that the #7-5 and #12-3 *CETN3*-KO hCOs were significantly smaller than control hCOs derived from the WT H9 cell line beginning at day 25 of culture (Fig. 2F), a stage that follows a period of NS/PC expansion and marks the beginning of neurogenesis. The reduced cerebral size phenotype observed in *CETN3*-deficient hCOs supported the hypothesis that *CETN3* disruption could be at least partially responsible for the reduced head circumference observed in the patient, and suggested a key role of CETN3 in brain development.

## Deletion of CETN3 affects the expression of genes involved in neuronal proliferation, differentiation, apoptosis, and splicing in cerebral organoids

To investigate the potential mechanisms underlying *CETN3* deletion-mediated microcephaly, we then conducted RNA-seq analysis of the H9 control and *CETN*-KO hCOs cultured for 45 days to identify genes or pathways potentially involved in the mechanisms, through which CETN3 might regulate brain development. Analysis of differentially expressed genes (DEGs)

identified 525 total DEGs (|log2FC| >0.5, *p*adj <0.05), including 191 up- and 334 down-regulated genes (Fig. 3A,B). Gene ontology (GO) enrichment analysis categorized the DEGs into four main groups: development, cell cycle, apoptosis, and splicing (Fig. 3C). The representative genes were shown in the heatmap (Fig. 3D). In the development group, the expression levels of NS/PC marker genes, including *PAX6* and *HES1* were decreased in *CETN3*-KO hCOs, while the expression levels of neuronal marker genes, *MAP2* and *DCX*, were significantly increased, indicating premature neuronal differentiation in the *CETN3*-KO hCOs (Fig. 3D). Consistently, the expression of *CHD5* and *ASCL1*, genes that promote neuronal differentiation (Shrestha et al, 2023; Vainorius et al, 2023), was upregulated, whereas *ID2*, a gene that negatively regulates cell differentiation (Peddada et al, 2006), was downregulated, suggesting that cell differentiation could be enhanced in *CETN3*-KO cells (Fig. 3D). In addition, the expression of genes known to promote neural progenitor proliferation, such as *PHOX2B* and *HMGA2* (Dubreuil et al, 2000; Nishino et al, 2008), was reduced, indicating insufficient maintenance of the NS/PC population in *CETN3*-KO groups compared to controls (Fig. 3D). Furthermore, the expression of genes related to cell cycle, apoptosis and RNA splicing, such as *CDC20*, *CHL1* and *CELF2* (Gao et al, 2021; Katic et al, 2017; Mallory et al, 2015) were changed in *CETN3*-KO hCOs, suggesting cell cycle, apoptosis and RNA splicing were all affected by *CETN3* deletion (Fig. 3D). These findings demonstrated that CETN3 regulates brain development and neurogenesis through multiple mechanisms. Further KEGG pathway analysis revealed that upregulated genes were enriched in pathways related to differentiation, including Hedgehog signaling and synapse formation, indicating enhanced neuronal differentiation in *CETN3*-KO hCOs (Fig. 3E). Conversely, downregulated genes were associated with pathways related to cell proliferation, such as MAPK signaling, Notch signaling, and Hippo signaling, suggesting that the proliferation of NS/PCs might be impaired in *CETN3*-KO hCOs (Fig. 3E).

## Deficiency of CETN3 interferes with neuronal differentiation and decreases NS/PC cell populations

To validate the above RNA-seq findings that suggest CETN3 could regulate brain development by interfering with cell proliferation, differentiation, and apoptosis, we collected H9 and *CETN3*-KO hCOs at day 45 for IF staining in cryosections. NS/PCs were labeled with antibodies for PAX6 or SOX2, while neurons of the human cortex were stained with antibodies for TBR1, CTIP2, and HuC/D, respectively (Fig. 4A–C) (Hsueh et al, 2000; Qu et al, 2023). The

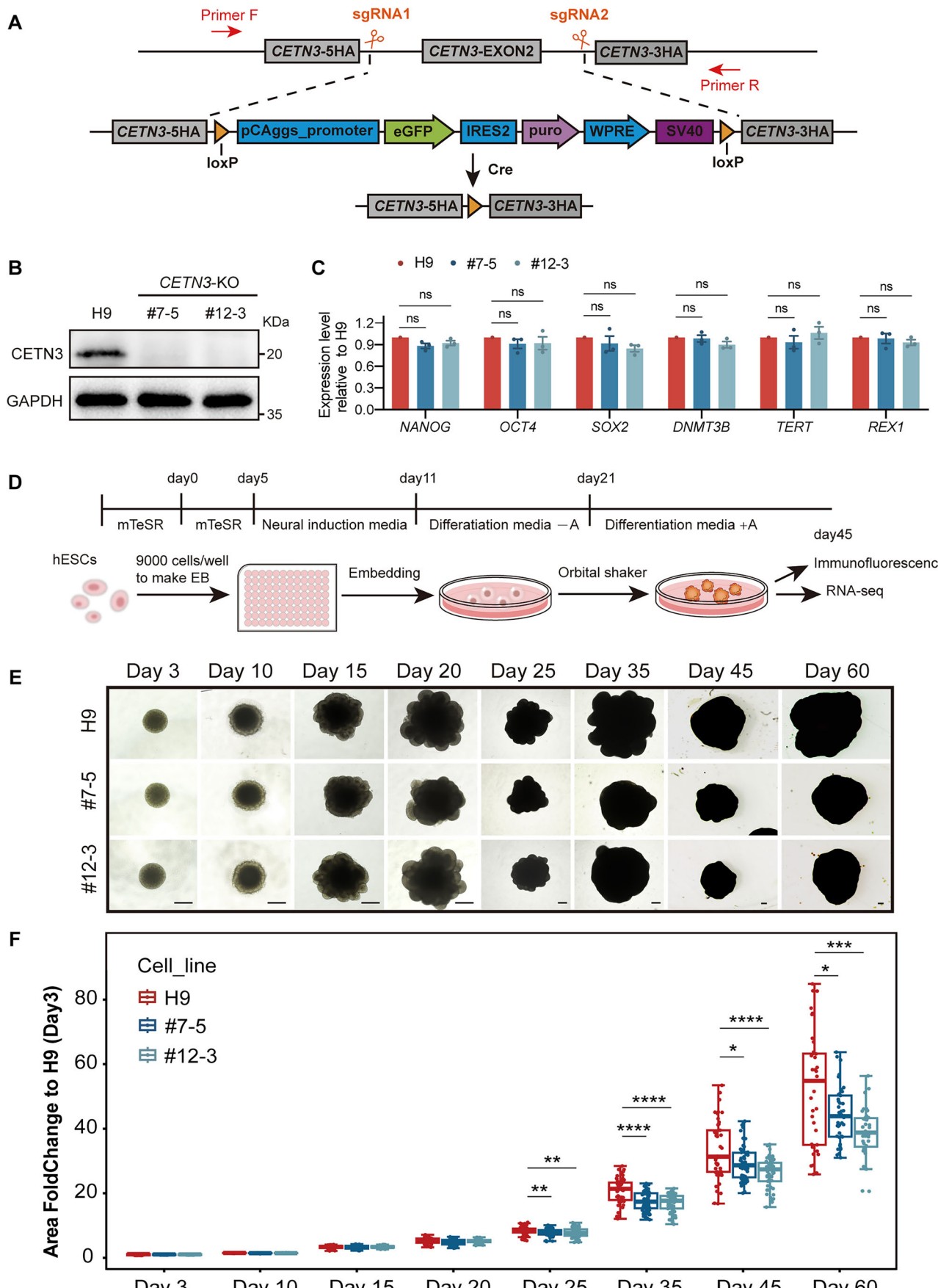

**Figure 2.  Deletion of CETN3 results in reduced sizes of human cerebral organoids.**

(A) The strategy for generating *CETN3*-KO cell lines based on the H9 hESC line using the CRISPR-Cas9 technique involved two sgRNAs targeting intron 1–2 and intron 2–3 to cleave the genomic DNA. Exon 2 of *CETN3* was replaced with an exogenous DNA fragment containing selection elements flanked by loxP sites. Cre recombinase was subsequently used to delete the exogenous sequence, thereby removing exon 2 of *CETN3*. Primers F and R were used for genotyping. (B) Western blot to confirm CETN3 deletion in #7-5 and #12-3 cell lines. (C) Bar plots showing qPCR analysis of pluripotency gene expression in H9, *CETN3*-KO clone #7-5, and clone #12-3. Results were normalized to H9 and presented as mean ± SEM. Statistical analysis was conducted using one-way ANOVA. ns not significant. (D) Schematic diagram for cerebral organoid culture. (E) Morphological and size analysis of organoids during culture. Scale bar: 400 μm. (F) Quantification of organoid size at different culture time points. Data were collected from four independent batches and normalized to H9 (day 3). Each plot represents one organoid (day 3: H9, $n = 56$, #7-5, $n = 54$, #12-3, $n = 52$; day 10: H9, $n = 53$, #7-5, $n = 52$, #12-3, $n = 54$; day 15: H9, $n = 66$, #7-5, $n = 67$, #12-3, $n = 71$; day 20: H9, $n = 60$, #7-5, $n = 62$, #12-3, $n = 62$; day 25: H9, $n = 67$, #7-5, $n = 71$, #12-3, $n = 71$; day 35: H9, $n = 59$, #7-5, $n = 72$, #12-3, $n = 71$; day 45: H9, $n = 45$, #7-5, $n = 57$, #12-3, $n = 53$; day 60: H9, $n = 37$, #7-5, $n = 42$, #12-3, $n = 40$). Boxes represent the interquartile range (IQR) from the first to third quartile, with the median shown as a horizontal line. Whiskers extend to the most extreme data points within 1.5× IQR; outliers beyond this range are shown as individual dots. Day 25: $P = 0.009$ (H9 vs. #7-5), $P = 0.005$ (H9 vs. #12-3); day 35: $P < 0.0001$ (H9 vs. #7-5), $P < 0.0001$ (H9 vs. #12-3); day 45: $P = 0.025$ (H9 vs. #7-5), $P < 0.0001$ (H9 vs. #12-3); day 60: $P = 0.038$ (H9 vs. #7-5), $P = 0.000201$ (H9 vs. #12-3). Statistical analysis was conducted using one-way ANOVA. $*P < 0.05$; $**P < 0.01$; $***P < 0.001$; $****P < 0.0001$. Source data are available online for this figure.

analysis revealed that the ratios of TBR1⁺, CTIP2⁺ or HuC/D⁺ neurons relative to PAX6⁺ NS/PCs were significantly increased in *CETN3*-KO hCOs compared to those in WT H9 hCOs (Fig. 4H–J). Additionally, staining for TBR2⁺ intermediate progenitor cells indicated that this population was also significantly expanded in *CETN3*-KO hCOs (Fig. 4D,K), collectively suggesting that CETN3 deficiency could lead to premature neuronal differentiation. IF staining for mitotic marker PH3 and cell cycle marker Ki67 to examine cell proliferation dynamics (Fig. 4E,F) showed a decrease in proliferating cells following loss of CETN3 (Fig. 4L,M), whereas staining for cleaved caspase-3 indicated that the proportion of apoptotic NS/PCs was elevated in *CETN3*-KO hCOs (Fig. 4G,N). To assess the differentiation and proliferation of NS/PCs at an earlier developmental stage, we also performed IF staining on hCOs at day 35, when neurons begin to emerge (Appendix Fig. S3A–C). The results were consistent with those observed in hCOs at day 45 (Appendix Fig. S3D–F). These cumulative results supported our RNA-seq analysis, and demonstrated that loss of CETN3 led to reduced size of hCOs by promoting neuronal differentiation, decreasing NS/PC proliferation, and inducing apoptosis.

To strengthen the reliability of our phenotypic findings and eliminate potential variations due to different cell lines, we additionally generated hCOs from both H1 hESCs and iPSCs. Specifically, *CETN3* was knocked out in H1 cells using the same approach employed for generating *CETN3*-KO H9 cell lines (Fig. 2A). The biallelic mutations identified in the *CETN3* gene of the patient were introduced into WT iPSCs using two recombinant plasmids (Appendix Fig. S4A). Successful introduction of these mutations was confirmed by Sanger sequencing (Appendix Fig. S4B). Western blot analysis demonstrated a complete loss of CETN3 protein in both *CETN3*-KO H1 cells and *CETN3*-mutant iPSCs (Appendix Fig. S4C,D). Using these two genetically modified cell lines, we generated hCOs, both of which exhibited reduced size compared to their respective controls (Figs. EV2A,B and EV3A,B), recapitulating the phenotype observed in *CETN3*-KO H9-derived hCOs. To further investigate cell differentiation, proliferation, and apoptosis, we collected hCOs at day 45 and performed IF staining. NS/PCs were identified by PAX6 or SOX2, while differentiated neurons were labeled with TBR1 or TUJ1. Cell proliferation was assessed by PH3 and Ki67 staining, and cell apoptosis was evaluated by cleaved caspase-3 staining (Figs. EV2C–G and EV3C–G). The results were consistent with those obtained from H9-derived hCOs, showing enhanced neuronal differentiation,

reduced NS/PC proliferation, and increased NS/PC apoptosis (Figs. EV2H–L and EV3H–L). Together, these findings supported our conclusion that CETN3 ablation led to reduced hCO size by promoting neuronal differentiation, inhibiting NS/PC proliferation, and inducing NS/PC apoptosis.

## CETN3 deficiency impacts mitotic spindle orientation, cell cycle dynamics, and spindle structure in NS/PCs

As CETN3 has been previously identified as a component of centrosomes (Dantas et al, 2011; Sawant et al, 2015), and transcriptomic profiling showed enrichment for cell cycle genes, we thus focused on cell division and cell cycling pathways to explore the cellular mechanisms of cell proliferation, differentiation, and apoptosis regulation by CETN3. Since the pool of early NS/PCs is initially expanded through symmetric division (vertical cleavage), then shifts to asymmetric division (oblique and horizontal cleavage), giving rise to one NS/PC daughter cell and one differentiating cell, coordinated by mitotic spindle orientation at the onset of neurogenesis (Lancaster and Knoblich, 2012; Paridaen and Huttner, 2014), we quantified mitotic cells and measured their splitting angles on the apical surface of the VZ in hCOs. We labeled mitotic cells with an antibody against PH3 and p-vimentin, then selected only cells in anaphase and telophase for analysis. Measurement of the angle between the cleavage plane and apical surface, and the complementary angle, which represents spindle orientation (Fig. 5A), revealed that 21% of NS/PCs in #7-5 hCOs and 37% of NS/PCs in #12-3 hCOs had spindle angle >30°, compared to 12% of NS/PCs in H9 hCOs (Fig. 5B), indicating preferential asymmetric division among NS/PCs in *CETN3*-KO hCOs compared to that in controls. We therefore hypothesized that a shift towards asymmetric division might contribute to the premature neuronal differentiation observed in *CETN3*-KO hCOs.

To further investigate NS/PC proliferation and division, we used 2D differentiation based on the Dual-SMAD inhibition method (Fig. EV4A) (Chambers et al, 2009) to generate NS/PCs from hESCs. IF detection of the NS/PC markers, PAX6 and SOX1, and neuronal marker, TUJ1, confirmed the successful generation of high percentage of NS/PC populations in the two KO and WT H9 cell lines (Fig. EV4B). Subsequent assessment of cell proliferation by staining for Ki67 and PH3 indicated that the proportion of PH3⁺ cells was significantly decreased in *CETN3*-KO NS/PCs compared to that in control cells (Fig. EV4C–E). Further quantification of the

**A**

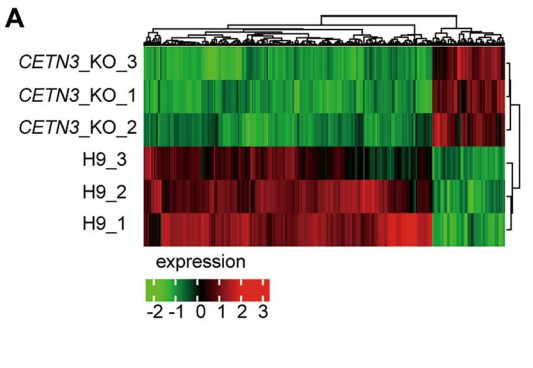

**B**

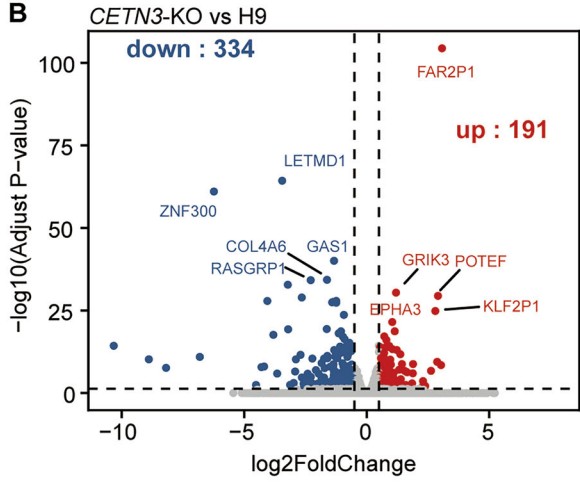

**D**

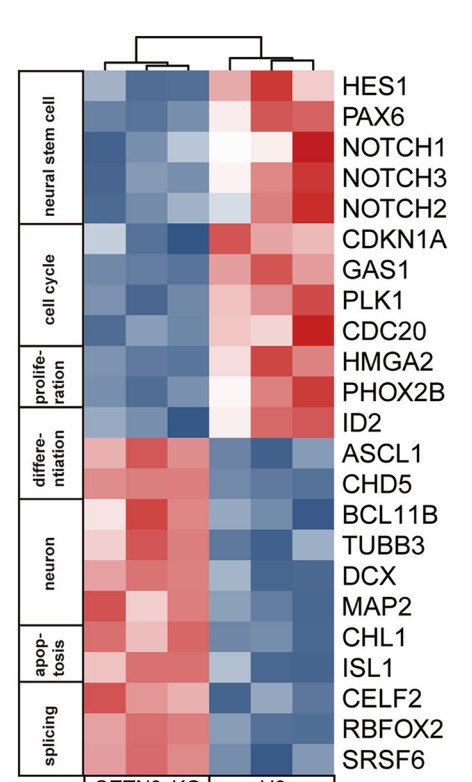

**C**

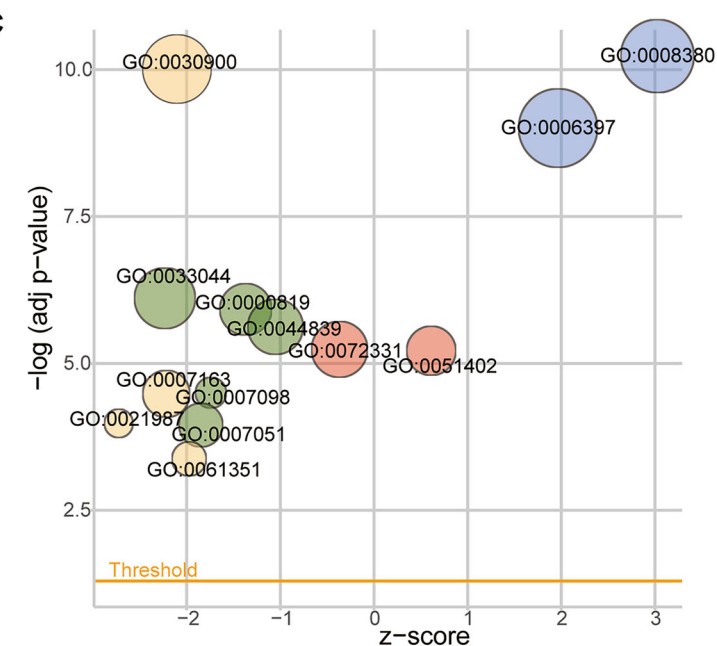

| ID | Description | |
|---|---|---|
| GO:0030900 | forebrain development (362) | development |
| GO:0007163 | establishment or maintenance of cell polarity (200) | development |
| GO:0021987 | cerebral cortex development (110) | development |
| GO:0061351 | neural precursor cell proliferation (133) | development |
| GO:0033044 | regulation of chromosome organization (305) | cell cycle |
| GO:0000819 | sister chromatid segregation (235) | cell cycle |
| GO:0044839 | cell cycle G2/M phase transition (261) | cell cycle |
| GO:0007098 | centrosome cycle (119) | cell cycle |
| GO:0007051 | spindle organization (183) | cell cycle |
| GO:0072331 | signal transduction by p53 class mediator (266) | apoptosis |
| GO:0051402 | neuron apoptotic process (217) | apoptosis |
| GO:0008380 | RNA splicing (420) | splicing |
| GO:0006397 | mRNA processing (479) | splicing |

**E**

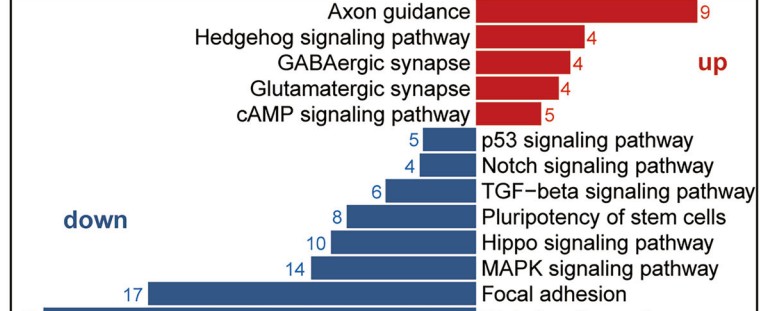

◀ **Figure 3.  Deletion of CETN3 affects the expression of genes involved in neuronal proliferation, differentiation, apoptosis, and splicing in cerebral organoids.**

(A) Heatmap showed differentially expressed genes (DEGs) of H9 and *CETN3*-KO organoids at day 45, based on RNA sequencing data. Each group included three biological replicates, with each replicate comprising three organoids. (B) Volcano plot depicted DEGs (*padj* <0.05, |log2FC| >0.5) between H9 and *CETN3*-KO organoids. The top five genes were labeled on the plot. Statistical significance of differential gene expression was determined in DESeq2 using a Wald test, with *p* values adjusted for multiple comparisons by the Benjamini–Hochberg procedure. (C) GO term enrichment analysis of DEGs. GO_MF enrichment bubble plot (upper), and description for the GO terms (lower). z-score does not refer to the standard score from statistics but is an easy to calculate value to give us a hint if the GO term is more likely to be decreased (negative value) or increased (positive value). $zscore = (up - down)/\sqrt{count}$. The GO terms were manually classified into four categories according to their biological functions, and gene counts for each term were labeled behind. (D) Heatmap of DEGs relevant to GO terms in panel (C). The genes were artificially classified into seven sets according to their functions. (E) KEGG signaling pathway enrichment analysis of DEGs. The numbers beside each bar were gene counts for each pathway. Red, upregulated pathway; blue, downregulated pathway. Enrichment analysis was performed using clusterProfiler to identify GO terms and KEGG pathways enriched among differentially expressed genes. Gene set *P* values were calculated using a hypergeometric test and adjusted for multiple comparisons using the Benjamini–Hochberg method.

percentages of ESCs and NS/PCs in the G1, S, and G2/M phases by flow cytometry using DAPI and EdU, followed by PH3 to distinguish the G2 and M phases (Fig. 5C), showed no significant differences in the proportion of each phase between H9 control ESCs and *CETN3*-KO ESCs (Fig. 5C). However, the proportion in S phase of *CETN3*-KO NS/PCs was significantly lower than that in H9 controls (Fig. 5D), while a larger proportion of *CETN3*-KO NS/PCs were detected in G2/M (Fig. 5E) with a notable reduction in the M phase compared to H9 controls (Fig. 5F). These results suggested that G1/S and G2/M transitions were impaired in the absence of CETN3.

We next explored how loss of CETN3 could affect cell cycle transitions in NS/PCs. CETN3 has been recently identified as a potential interaction partner of USP44 (Oh et al, 2020), a deubiquitinase targeting the cell-cycle regulator CDC20 (Stegmeier et al, 2007; Zhang et al, 2011). We conducted a co-immunoprecipitation (CoIP) assay to verify the interaction between CETN3 and USP44 in HEK 293T cells (Fig. 5G). Previous work has shown that USP44 could regulate the spindle assembly checkpoint by preventing early anaphase onset via deubiquitinating CDC20, thereby suppressing the activity of Cdc20-anaphase promoting complex/cyclosome (Cdc20-APC/C) E3 ubiquitin ligase (Stegmeier et al, 2007; Zhang et al, 2011). We thus hypothesized that loss of CETN3 in NS/PCs could influence Cdc20-APC/C activity by affecting the function of USP44 during mitosis. To test this hypothesis, we assessed the levels of CyclinB1, a known substrate of Cdc20-APC/C, via immunoblotting. NS/PCs derived from 2D differentiations were treated with the microtubule polymerization inhibitor nocodazole for 24 h to arrest cells in mitosis, followed by a 1-hour release. Cyclin B1 levels were elevated in *CETN3*-KO cells compared to H9 controls, suggesting a reduction in Cdc20-APC/C activity (Fig. 5H,I). Additionally, increased levels of PH3 indicated that fewer cells completed mitosis in *CETN3*-KO cells (Fig. 5H,J), likely due to diminished Cdc20-APC/C activity. These findings suggested that the loss of CETN3 impaired APC/C activity, possibly by enhancing USP44-mediated deubiquitination of CDC20.

Finally, CETN3-deficient NS/PCs more frequently showed aberrant spindles compared to H9-derived NS/PCs (Fig. 5K,L; H9: 2.84%, n = 282; #7-5: 9.85%, n = 203; #12-3: 9.23%, n = 325), which might be responsible for increased cell apoptosis in CETN3-deficient NS/PCs (Fig. EV4F). In summary, the proliferative capacity of *CETN3*-KO NS/PCs is diminished due to alterations in the cell cycle, and the increased apoptosis is possibly due to abnormal centrosome duplication and defective cell division.

## CETN3 regulates RNA splicing of centriole-related genes, indirectly influencing cell cycle

In addition to the roles of CETN3 on cell cycle in balancing NS/PC self-renewal and neuronal differentiation, we also noted that pathways related to RNA splicing were significantly enriched in *CETN3*-KO hCOs. To explore whether and how such potential CETN3 functions in regulating RNA splicing might contribute to NS/PC alterations, we next conducted RNA-seq analysis of *CETN3*-KO and WT H9 NS/PCs derived using the 2D differentiation approach. We identified 2562 DEGs between KO versus H9 NS/PCs (Fig. EV5A,B). GO terms were enriched in development, cell cycle, apoptosis, and RNA splicing processes (Fig. EV5C), which were similar to those identified in transcriptomic profiling from hCOs (Fig. 3C), and largely involved the similar pathways to those enriched in hCOs (Fig. EV5D). Similarly, RNA splicing GO term were also enriched in NS/PCs. To identify possible alterations in RNA splicing, using rMATS software, we identify 848 and 1723 differentially spliced genes in hCOs and 2D-differentiated NS/PCs, respectively. Among those genes, 280 genes overlapped between hCOs and 2D-differentiated NS/PCs (Fig. EV5E). The higher number of ASEs in NS/PCs compared to hCOs suggested that CETN3 might exhibit greater impacts on RNA splicing in NS/PCs, which led us to focus on NS/PCs in subsequent analyses.

Classification (Fig. 6A) of significantly altered alternative splicing events (ASEs) under *CETN3* KO identified 1725 significant skipped exon (SE) events, 124 alternative 5′ splice site (A5SS) events, 215 alternative 3′ splice site (A3SS) events, 182 mutually exclusive exons (MXE), and 127 retained intron (RI) events (Fig. 6B–F). GO term analysis of genes affected by alternative splicing revealed enrichment in functional annotations related to spindle, centriole, cilium, and cell cycle (Fig. 6G), while KEGG pathway analysis showed enrichment in the MAPK and Wnt signaling pathways (Fig. 6H). Among these differentially spliced genes, we selected two representative genes related to spindle, centriole and ciliogenesis, *PCM1* and *CEP250*, and two representative genes relevant to neural development, *TENM4* and *CADM3*, for RT-PCR validation (Appendix Table S1), which corroborated the peaks in IGV tracks (Fig. 6I,J; Appendix Fig. S5A,B). To further verify the effects of CETN3 on RNA splicing of ciliogenesis-related genes, we conducted IF staining to detect ARL13b that showed significant decreased cilia density in CETN3-deficient hCOs (Appendix Fig. S5C,D).

To define the molecular mechanisms of CETN3 on RNA splicing during neural development, we looked up in the BioGRID

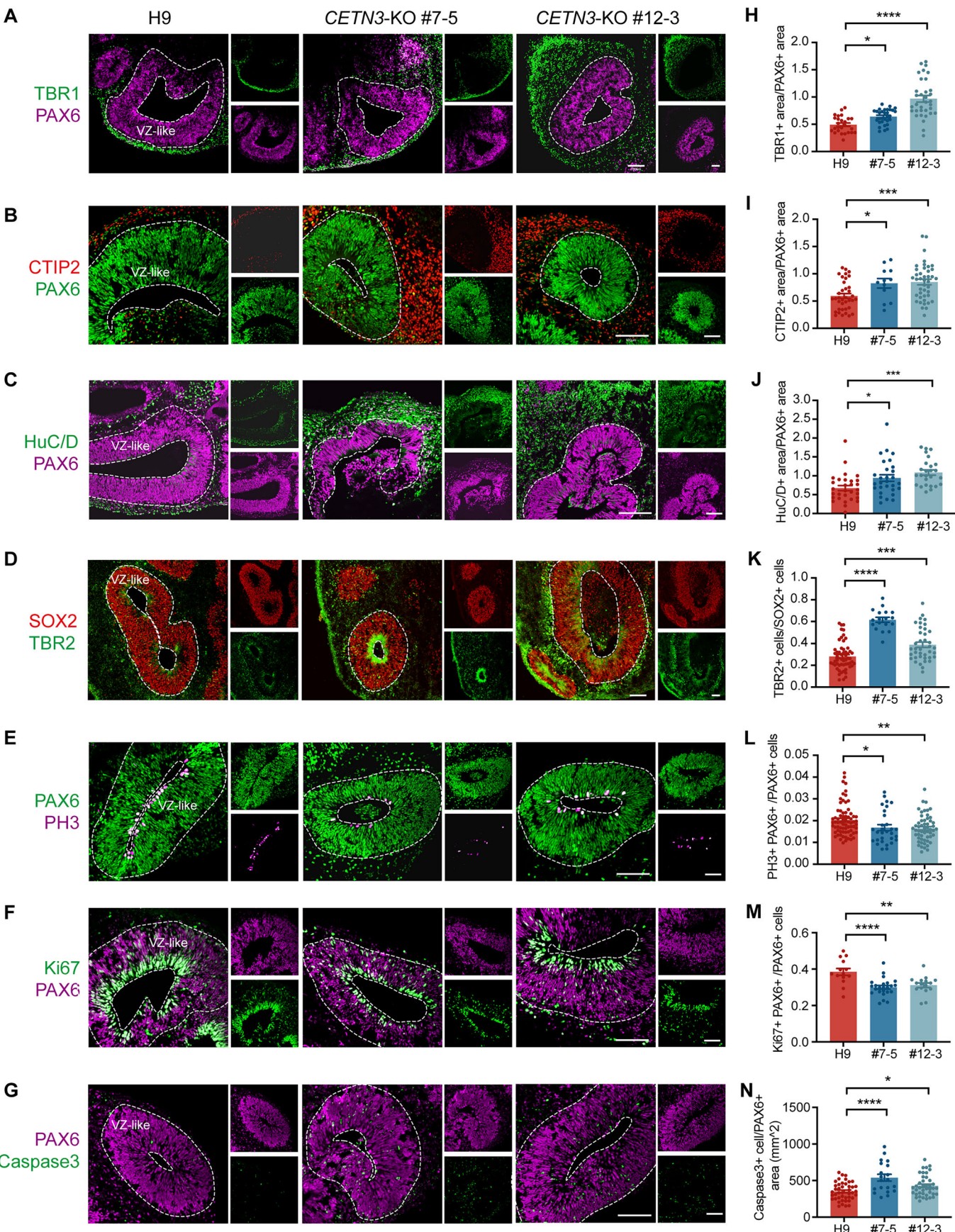

**Figure 4. Deficiency of CETN3 interferes with neuronal differentiation and diminishes the maintenance of NS/PCs.**

(A–C) Immunofluorescence staining of cryosections from day 45 cerebral organoids. NS/PCs were labeled with PAX6, while neurons were identified using TBR1, CTIP2, and HuC/D. Scale bar: 100 μm. VZ ventricular zone. (D) Immunofluorescence staining of cryosections from day 60 cerebral organoids. NS/PCs were labeled with SOX2, and intermediate progenitors were identified using TBR2. Scale bar: 100 μm. (E–G) Immunofluorescence staining of cryosections from day 45 cerebral organoids. Markers used included PAX6 for NS/PCs, PH3 (E) and Ki67 (F) for cell proliferation, and cleaved caspase-3 (G) for apoptosis. Scale bar: 100 μm. (H–N) Quantification of cell number or staining area for each marker. Each dot represented an individual rosette or several adjacent rosettes (H: H9, $n = 26$, #7-5, $n = 31$, #12-3, $n = 34$; I H9, $n = 37$, #7-5, $n = 12$, #12-3, $n = 42$; J: H9, $n = 30$, #7-5, $n = 29$, #12-3, $n = 26$; K: H9, $n = 56$, #7-5, $n = 18$, #12-3, $n = 39$; L: H9, $n = 73$, #7-5, $n = 30$, #12-3, $n = 56$; M: H9, $n = 14$, #7-5, $n = 24$, #12-3, $n = 16$; N: H9, $n = 43$, #7-5, $n = 20$, #12-3, $n = 41$). Data were collected from organoids across three independent experiments, with results presented as mean ± SEM. (H) $P = 0.0424$ (H9 vs. #7-5), $P < 0.0001$ (H9 vs. #12-3); (I) $P = 0.0454$ (H9 vs. #7-5), $P = 0.0007$ (H9 vs. #12-3); (J) $P = 0.0247$ (H9 vs. #7-5), $P = 0.0005$ (H9 vs. #12-3); (K) $P < 0.0001$ (H9 vs. #7-5), $P = 0.0003$ (H9 vs. #12-3); (L) $P = 0.0325$ (H9 vs. #7-5), $P = 0.006$ (H9 vs. #12-3); (M) $P < 0.0001$ (H9 vs. #7-5), $P = 0.0014$ (H9 vs. #12-3); (N) $P < 0.0001$ (H9 vs. #7-5), $P = 0.0426$ (H9 vs. #12-3). Differential analysis was performed using one-way ANOVA. $*P < 0.05$; $**P < 0.01$; $***P < 0.001$; $****P < 0.0001$. Source data are available online for this figure.

(https://thebiogrid.org/) database to search for potential CETN3-binding proteins, which identified USP49, a deubiquitinase that regulates co-transcriptional pre-mRNA splicing by deubiquitinating histone H2B (Tu et al, 2022; Zhang et al, 2013). Subsequent CoIP in HEK 293T cells confirmed that CETN3 could interact with USP49 (Fig. 6K). Furthermore, the loss of CETN3 significantly impaired the deubiquitinase activity of USP49 on H2B in 2D-differentiated NS/PCs (Fig. 6L–N). These cumulative findings supported the possibility that CETN3 may participate in USP49-mediated RNA splicing, and suggested both direct and indirect roles in regulating cell cycle and splicing of genes involved in cell cycle regulation in neural development.

## Discussion

Microcephaly is a relatively rare condition, with incidence ranging from 0.001 to 0.15% (Becerra-Solano et al, 2021; Messinger et al, 2020; Morris et al, 2016; Nunez et al, 2021). Primary microcephaly is largely attributed to inadequate neuron production during neurogenesis. In this study, we present a clinical case of primary microcephaly resulting from biallelic loss-of-function mutations in *CETN3*, inherited from the parents. According to previous studies, CETN3 is a centrin protein known to play critical roles in the microtubule-organizing center and mRNA export to the cytoplasm through nuclear pores. However, its specific functions in brain development and neurogenesis have not been reported. To validate the pathogenicity of CETN3 loss and investigate its role in brain development, we generated *CETN3*-KO hESC and iPSC lines, which were also used to generate hCOs and 2D-differentiated NS/PCs. Findings obtained through RNA-seq, IF, and flow cytometry demonstrated that CETN3 influences neurogenesis by modulating NS/PC proliferation, differentiation, and apoptosis. We observed that NS/PCs with CETN3 deletion exhibit accelerated differentiation into neurons due to altered mitotic spindle orientation. CETN3 deficiency also leads to impaired NS/PC proliferation through impaired transitions in the cell cycle, and increased apoptosis. Unexpectedly, RNA-seq and CoIP analyses suggested that CETN3 could also indirectly affect NS/PC fate determination through interactions with a deubiquitinase involved in RNA splicing, USP49, which regulated alternative splicing of genes relevant to centriole and cilia. These results highlight the novel role of CETN3 in controlling the balance between NS/PC proliferation and differentiation required for normal brain development.

This study thus provides, to our knowledge, the first causal evidence linking *CETN3* to microcephaly due to reduced neural

progenitor populations. The neocortex, a brain structure unique to mammals, is where neurogenesis occurs most intensively, and its size generally correlates with overall brain size in mammals (Florio and Huttner, 2014). During neurogenesis in the neocortex, NS/PCs undergo either symmetric division, producing two NS/PCs, or asymmetric division, producing a NS/PC and a differentiating daughter cell. The right number of neurons depends on the sufficient NS/PCs at the early stage of neurogenesis (Florio and Huttner, 2014; Kang et al, 2017; Mérot et al, 2009; Paridaen and Huttner, 2014). Given the importance of the NS/PC pool, it is unsurprising that many MCPH genes are involved in cell cycle, centrosome, DNA repair, and apoptosis, all of which affect NS/PC numbers. Our findings suggest that the loss of CETN3, a centrosome protein, contributes to microcephaly by disrupting the regulation of NS/PC behavior during neurogenesis.

Importantly, our findings uncovered a long-debated role of CETN3 in centrosome duplication. While the loss of CETN3 did not affect cell cycle or proliferation in hESCs, consistent with the results of Dantas et al in DT40 cells (Dantas et al, 2011), we observed the emergence of a few aberrant spindles (about 1.5%) in CETN3-deficient hESCs. The failure to detect a mitotic phenotype in *CETN3*-overexpressing HeLa cells (Middendorp et al, 2000) may have been due to the low incidence of such abnormalities, as we only identified spindle defects after analyzing hundreds of cells. Moreover, our findings are in line with another study that reported CETN3 depletion promoted centriole reduplication in Hela cells (Sawant et al, 2015), although the frequency of such multi-polar spindles substantially differed between studies (i.e., ~1% in our experiment compared to >50% in Sawant et al), which might be due to different experimental conditions. Moreover, we found that the proportion of aberrant spindles increased to ~10% in *CETN3*-KO NS/PCs. Unlike in hESCs, CETN3 deficiency in NS/PCs also affected cell cycle, proliferation, and apoptosis. These observations suggested a cell type-specific function of CETN3, highlighting its distinct and context-dependent roles.

It is also noteworthy that our study uncovered a previously unrecognized function of CETN3 in RNA splicing via interacting with USP49 and facilitating its deubiquitinase activity on histone H2B. We found that different ASEs were enriched in processes associated with centriole, cell cycle, cilium, etc., implying that CETN3 could indirectly regulate cell cycle, mitosis, and ciliogenesis. The effects of *CETN3* KO on alternative splicing was validated by RT-qPCR assays of two centriole-related genes (Fig. 6I,J) and two neural development-related genes (Appendix Fig. S5A,B). These findings thus expand our understanding of CETN3's

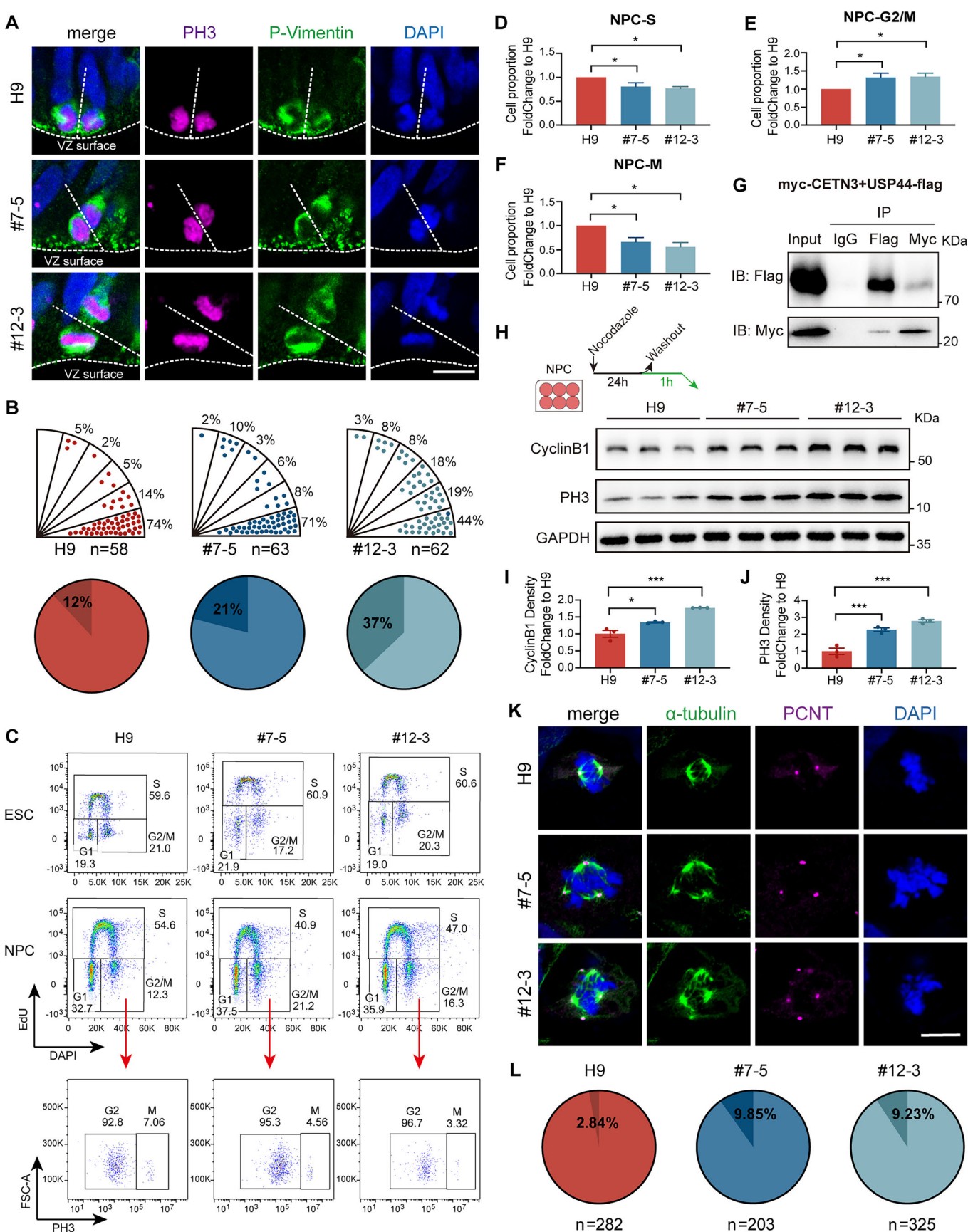

**Figure 5. CETN3 deficiency impacts mitotic spindle orientation, cell cycle dynamics, and spindle structure in NS/PCs.**

(A) Immunofluorescence for day 45 cerebral organoids cryosections. Markers for mitosis, PH3 and P-Vimentin, were stained. Representative cells on the apical surface undergoing mitosis were magnified. The dashed line indicates the apical surface and mitotic splitting plane. Scale bar: 10 μm. (B) Distribution of spindle orientations relative to the apical surface in H9 and two *CETN3*-KO cell lines (upper). Each dot represented an individual cell, with proportions of cells in each orientation interval indicated. The organoids used for staining statistics were collected from three independent experiments. Cell proportions with spindle angles greater than 30° were illustrated in the pie charts (lower). Red: H9; blue: #7-5; green: #12-3. The darker sections in each pie chart indicated cells with spindle angles greater than 30°, with the numbers representing their respective proportions. (C) Cell cycle analysis of ESCs and NS/PCs using flow cytometry. ESCs were categorized into G1, S, and G2/M phases based on DAPI and EdU staining (upper). NS/PCs were similarly divided into G1, S, and G2/M phases using DAPI and EdU staining (middle), with G2 and M phases further distinguished by PH3 signal (lower). Proportions of cells in each phase were indicated on the plot. EdU labeling time was 0.5 h for ESCs and 3 h for NS/PCs. (D–F) Comparison of NS/PC proportions across different cell lines in S (D), G2/M (E), and M (F) phases. The data were collected from six batches for S and G2/M phases, three batches for M phase, and all the data were normalized to H9. Data were shown as mean ± SEM. (D) $P = 0.0315$ (H9 vs. #7-5), $P = 0.0101$ (H9 vs. #12-3); (E) $P = 0.0396$ (H9 vs. #7-5), $P = 0.0258$ (H9 vs. #12-3); (F) $P = 0.0281$ (H9 vs. #7-5), $P = 0.0148$ (H9 vs. #12-3). One-way ANOVA was used for differential analysis. $^*P < 0.05$. (G) CoIP to certify the interaction of CETN3 and USP44. Myc and flag tags were respectively added to the N-terminal and C-terminal of CETN3 and USP44. (H) Western blot to assess the levels of CyclinB1 and PH3 in 2D-differentiated NS/PCs. GAPDH was used as a reference. Three replicates for each group. NS/PCs were treated with nocodazole for 24 hours, followed by a 1-h release. (I, J) Western blot quantification for CyclinB1 (I) and PH3 (J). Data were presented as the mean ± SEM, calculated from three replicates and normalized to H9. (I) $P = 0.0138$ (H9 vs. #7-5), $P = 0.0002$ (H9 vs. #12-3); (J) $P = 0.0010$ (H9 vs. #7-5), $P = 0.0002$ (H9 vs. #12-3). One-way ANOVA was used for differential analysis. $^*P < 0.05$; $^{**}P < 0.001$. (K) Immunofluorescence staining of NS/PCs from 2D differentiation. Markers for microtubule, α-tubulin, for centrosome, PCNT, were stained. Representative cells undergoing mitosis were magnified for each cell line. Scale bar: 10 μm. (L) Statistics for the proportion of aberrant spindles in each cell line. Red: H9; blue: #7-5; green: #12-3. The darker sections in each pie chart indicated aberrant spindles, with the numbers representing their respective proportions. Data were collected from three independent experiments. Source data are available online for this figure.

contributions to mitosis and ciliogenesis during neural development. Previous studies have indicated that anomalies in the splicing system are associated with microcephaly (Pan et al, 2023; Patel et al, 2024; Yang et al, 2024). For example, Chai et al demonstrated the pathogenicity of *PPIL1* and *PRP17*, two RNA splicing-related genes, in patients with pontocerebellar hypoplasia and microcephaly (Chai et al, 2021). Additionally, the deletion of *Argku1*, a pre-mRNA splicing regulator, in the mouse cortex resulted in impaired neurogenesis and microcephaly (Yao et al, 2023). Furthermore, inactivation of the minor spliceosome in the developing mouse cortex caused microcephaly through disruptions in the cell cycle and induced neural stem cell death (Baumgartner et al, 2018). Consistent with these findings, our results show that genes with altered RNA splicing in the *CETN3*-KO group were enriched in cell-cycle-, spindle-, and centrosome-related GO terms, suggesting that changes in RNA splicing may contribute to the microcephaly phenotype observed in our patient. Nevertheless, further detailed study will be necessary to examine the specific pathways, genes, and mechanisms disrupted by aberrant mRNA splicing due to CETN3 loss of function that are potentially involved in the pathogenesis of microcephaly.

One study investigated the role of CETN3 in an in vivo mouse model and reported that deletion of *Cetn3* alone did not cause a discernible phenotype in retinal development. However, simultaneous ablation of *Cetn2* and *Cetn3* led to remarkably impaired retina ciliogenesis, suggesting that Cetn2 might compensate for the function of Cetn3 in maintaining cilium structure in mice (Ying et al, 2019). To further explore whether deletion of Cetn3 causes microcephalic phenotype in mice, we generated forebrain-specific *Cetn3*-ablated mice (*Cetn3^{folx/folx};Emx1-Cre*) by crossing *Cetn3^{flox/flox}* with *Cetn3^{folx/+};Emx1-Cre* mice, using littermates without *Emx1-Cre* as control (Fig. EV6A,B). Firstly, the deletion of Cetn3 in the mouse forebrain was confirmed by IF staining (Fig. EV6C). At E13.5, a significant but subtle reduction in forebrain size was detected in *Cetn3*-KO mice compared to control (Fig. EV6D,E), while no difference was observed at P0 (Fig. EV6F,G). The absence of a microcephaly phenotype in neonatal *Cetn3*-KO mice may be due to

compensation by *Cetn2*. However, some studies have indicated that while human CETN3 and CETN2 share similar functions, they are not redundant and cannot fully compensate for each other (Sawant et al, 2015; Vonderfecht et al, 2012). This species difference may account for the observed discrepancy in phenotypes between mice and humans. Further studies will be required to elucidate the functional redundancy and difference of centrins across different species.

In summary, this study provides, to our knowledge, the first report defining *CETN3* as a causal gene in microcephaly. Subsequent characterization of the effects of CETN3 loss of function in human cells and organoids suggests impairment of neurogenesis due to defects in NS/PC cycling caused by centriole/spindle dysfunction that might block differentiation and proliferation pathways required for embryonic brain development. Moreover, we found that CETN3 deficiency also leads to altered RNA splicing of genes involved in these neural development processes, thus suggesting an indirect regulatory mechanism in brain development.

# Methods

**Reagents and tools table**

| Reagent/resource | Reference or source | Identifier or catalog number |
|---|---|---|
| **Experimental models** | | |
| HEK-293 cells (*H. sapiens*) | ATCC | CRL-1573 |
| hESC H1 (*H. sapiens*) | WiCell | WA01 |
| hESC H9 (*H. sapiens*) | WiCell | WA09 |
| human iPSC (*H. sapiens*) | Nuwacell | ZSSY001 |
| C57BL6/J (*M. musculus*) *Cetn3^{flox/+}* | GemPharmatech | T021927 |
| C57BL6/J (*M. musculus*) *Emx1-Cre* | Shanghai Model Organisms Center | NM-KI-200149 |

| Reagent/resource | Reference or source | Identifier or catalog number |
|---|---|---|
| **Recombinant DNA** | | |
| pX458-Intron12-gRNA | This study | N/A |
| pX458-Intron23-gRNA | This study | N/A |
| pIVK33-CETN3-KO | This study | N/A |
| pIVK33-CETN3-mut1 | This study | N/A |
| pIVK33-CETN3-mut2 | This study | N/A |
| pCAG-myc-CETN3 | This study | N/A |
| pCAG-USP44-flag | This study | N/A |
| pCAG-USP49-flag | This study | N/A |
| pCAG-Cre | Addgene | #125574 |
| **Antibodies** | | |
| antibodies | This study | Appendix table S2 |
| **Oligonucleotides and other sequence-based reagents** | | |
| PCR primers | This study | Appendix table S1 |
| **Chemicals, enzymes and other reagents** | | |
| Matrigel Matrix hESC-Qualified | Corning | #354277 |
| Matrigel Basement Membrane Matrix | Corning | #354234 |
| mTeSR | Stemcell Technologies | #85850 |
| DMEM | Gibco | #C11995500BT |
| GlutaMax | Invitrogen | #35050-038 |
| penicillin/streptomycin | Sigma | #P0781 |
| polyethylenimine L (PEI) | Thermo Fisher | #BMS1003-A |
| ROCK inhibitor Y27632 | Selleckchem | #S1049 |
| DMEM/F12 | Invitrogen | #11330-032 |
| N2 Supplement | Invitrogen | #17502048 |
| MEM-NEAA | Sigma | #M7145 |
| Heparin | Sigma | #H3149 |
| Neurobasal | Invitrogen | #21103049 |
| B27 supplement without vitamin A | Invitrogen | #12587010 |
| B27 supplement with vitamin A | Invitrogen | #17504044 |
| insulin | Sigma | #I9278-5ML |
| Dorsomorphin | Selleckchem | #S1067 |
| SB431542 | Selleckchem | #S7840 |
| PrimeSTAR DNA Polymerase | Takara Bio | #R010A |
| 2× Taq Master Mix | Vazyme | #P112-01 |
| HiScript III RT SuperMix for qPCR | Vazyme | #R323-01 |
| Taq Pro Universal SYBR qPCR Master Mix | Vazyme | #Q712-02 |
| RIPA | Beyotime | #P0013B |
| DAPI | ThermoFisher | #D1306 |
| NaHCO$_3$ | Sigma | #S5761 |

| Reagent/resource | Reference or source | Identifier or catalog number |
|---|---|---|
| **Software** | | |
| fastp | https://github.com/OpenGene/fastp | |
| fastqc | https://www.bioinformatics.babraham.ac.uk/projects/fastqc/ | |
| Hisat2 v2.0.5 | https://daehwankimlab.github.io/hisat2/ | |
| featureCounts v1.5.0-p3 | https://subread.sourceforge.net/featureCounts.html | |
| DESeq2 v1.20.0 | https://bioconductor.org/packages/devel/bioc/vignettes/DESeq2/inst/doc/DESeq2.html | |
| clusterProfiler | https://bioconductor.org/packages/devel/bioc/html/clusterProfiler.html | |
| rMATS v4.2.0 | https://rnaseq-mats.sourceforge.io/download.html | |
| ImageJ | https://imagej.net/ij/index.html | |
| GraphPad Prism 8. | https://www.graphpad.com/features | |
| Primer 3 | https://bioinfo.ut.ee/primer3-0.4.0/ | |
| **Other** | | |
| Click-iT Edu Imaging Kit | Invitrogen | #C10338 |
| EndoFree Plasmid Maxi Kit | QIAGEN | #12243 |
| BCA Kit | Thermo Fisher | #A53225 |
| Confocal microscopy | Zeiss | LSM 880 |
| Flow cytometer | BD | FACSAria II |
| Cryostat Microtome | Leica | CM1950 |
| Neon™ NxT Electroporation System | Thermo Fisher | NEON1 |
| Neon consumables | Thermo Fisher | #MPK10096 |

## Genetic variant analysis

Whole-exome sequencing was performed on DNA samples from the patient and their parents. Exome enrichment utilized the Nextera Rapid Capture Exome v1.2 kit (Illumina), with sequencing conducted on the HiSeq4000 platform (Illumina), producing 150-bp paired-end reads. Data were analyzed using Berry Genomics' Verita Trekker Variant Detection System and the Enliven Variant Annotation and Interpretation System, referencing the human genome assembly hg19 (GRCh37). Variants not meeting quality control standards—depth of coverage below 10, allele balance under 0.25, or Phred quality score less than 20—were excluded. For comprehensive analysis, stringent criteria were applied: (1) heterozygous variants with a minor allele frequency (MAF) below 0.1% in databases such as 1000 Genomes, NHLBI-ESP, GnomAD, ExAC, and our in-house WES data; (2) homozygous or compound heterozygous variants with an MAF below 1% in these public databases; and (3) variants resulting in stop gain/loss, frameshift, splice-site, or missense changes. Besides *CETN3*, other potential pathogenic genes detected in the patient through this analysis were listed in Table EV1.

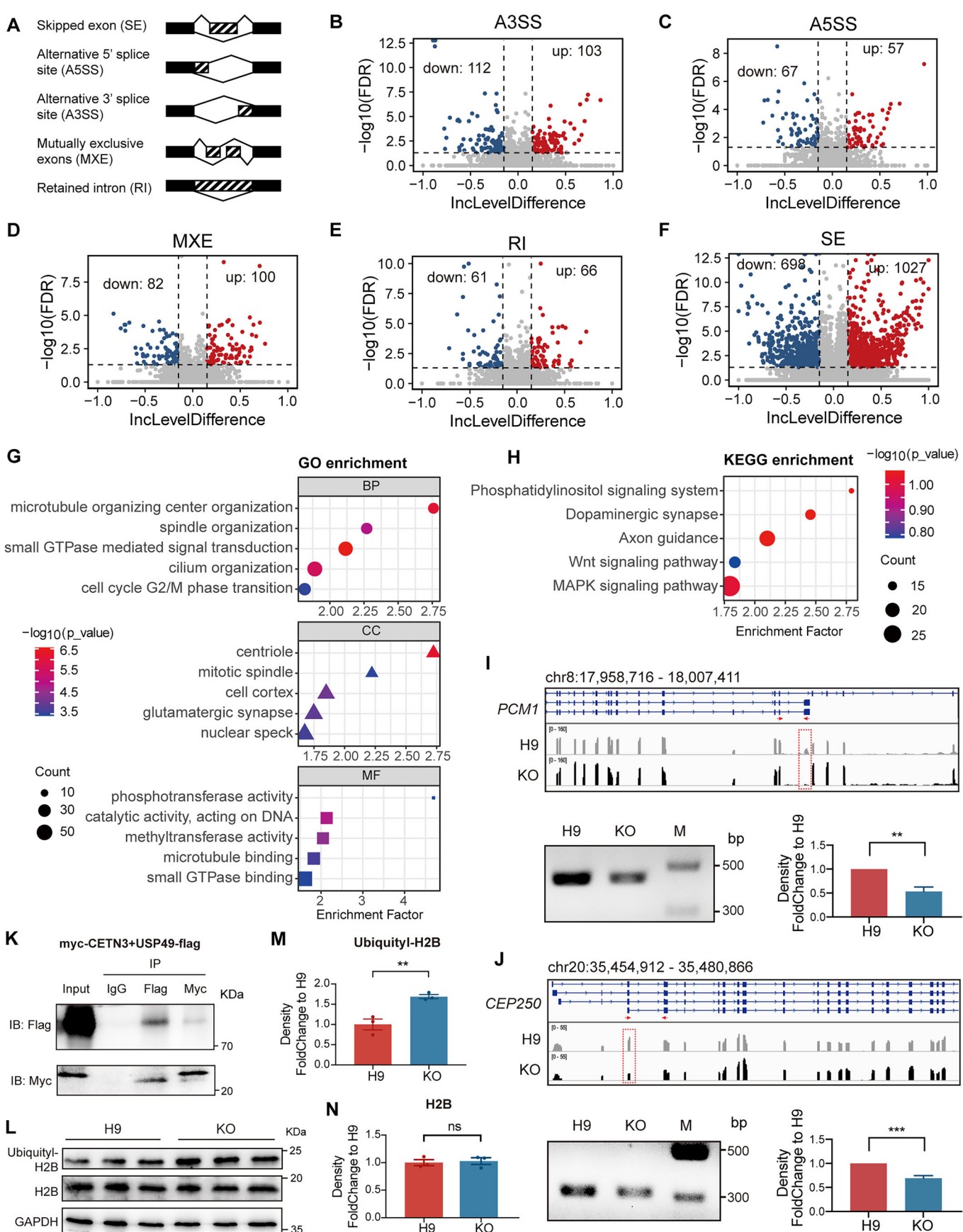

◄ **Figure 6. CETN3 exerts an indirect influence on the cell cycle by regulating the splicing of centriole-related genes.**

(A) Schematic diagram of five types of RNA splicing. Solid rectangles indicate constitutive exons, while dotted rectangles denote alternatively spliced exons. (B–F) Volcano plots illustrated up- and down-regulated splicing events in *CETN3*-KO compared to H9 for five types of RNA splicing. Data were from RNA-seq of NS/PCs derived from 2D differentiation. The numbers of significant up- and down-regulated events (FDR <0.05, IncLevelDifference >0.15) were labeled on each plot. Statistical significance of alternative splicing was assessed in rMATS using a likelihood ratio test based on a Bayesian model. Inclusion level difference (IncLevelDifference) was calculated as the difference in exon inclusion levels between groups. $P$ values were adjusted for multiple testing using the Benjamini–Hochberg method to obtain FDR. FDR false discovery rate calculated from $P$ value. IncLevelDifference: average (inclusion level for *CETN3*-KO) − average (inclusion level for H9). (G) GO enrichment results using different genes in splicing (*CETN3*-KO vs H9) as input. GO-BP, CC, and MF were plotted separately. Enrichment factor = GeneRatio/BgRatio. (H) KEGG enrichment results using different genes in splicing (*CETN3*-KO vs H9) as input. Enrichment analysis was performed using clusterProfiler to identify GO terms and KEGG pathways enriched among differentially expressed genes. Gene set $p$ values were calculated using a hypergeometric test and adjusted for multiple comparisons using the Benjamini–Hochberg method. (I, J) Splicing information for two representative centriole-related genes, *PCM1* and *CEP250*, in H9 and *CETN3*-KO NS/PCs. IGV tracks depicted the selected regions of target genes, with alternatively spliced exons highlighted by a red dashed frame (upper). Alternative splicing was validated by RT-PCR, with primers indicated by red arrows in the upper panel (lower left). Quantification of PCR results was shown as mean ± SEM, collected from three batches and normalized to H9 (lower right). (I) $P = 0.0017$ (H9 vs. KO); J: $P = 0.0002$ (H9 vs. KO). An unpaired $t$-test was used for differential analysis. **$P < 0.01$; ***$P < 0.001$. M marker. (K) CoIP to certify the interaction of CETN3 and USP49. Myc and flag tags were respectively added to the N-terminal and C-terminal of CETN3 and USP49. (L) Western blot to assess the levels of ubiquitinated H2B and H2B in 2D-differentiated NS/PCs. GAPDH was used as a reference. Three replicates for each group. (M, N) Western blot quantification for ubiquitinated H2B (M) and H2B (N). Data were presented as the mean ± SEM, calculated from three replicates and normalized to H9. (M) $P = 0.0086$ (H9 vs. KO); (N) $P = 0.7556$ (H9 vs. KO). An unpaired $t$-test was used for differential analysis. **$P < 0.01$; ns not significant. Source data are available online for this figure.

## Standard protocol approvals, registrations, and patient consents

Written informed consent was obtained from the participants, as approved by the Ethics Committee and the Expert Committee of Hunan Provincial Maternal and Child Health Care Hospital, University of South China.

## Plasmids

In this study, we employed a CRISPR-Cas9 knockout strategy to target the human *CETN3* gene. Initially, sgRNAs were designed using the online platforms Tefor (http://crispr.tefor.net) and CCTop (https://cctop.cos.uni-heidelberg.de:8043), and subsequently cloned into the pX458 vector (Addgene, #48138). Following an efficiency assessment in HEK 293T cells, two sgRNAs (5′-ACAGGTCATTGACTCATAAA-3′, 5′-GTTGTTTG CACCAGAGAAAG-3′) were selected, targeting *CETN3* intron 1–2 and intron 2–3, respectively, for the construction of the *CETN3*-KO cell line. For the homologous recombination donor plasmid, an eGFP-IRES-Puro expressing cassette flanked by loxp sequences and homology arms (HA), targeting *CETN3* intron 1–2 and intron 2–3, was cloned into pBlueScript SK(+) (Strata Gene). To test the interaction of CETN3 and USP44/49, human CETN3 and USP44/49 were cloned into pCAG-GFP vector (Addgene, #11150). Primers used for plasmid construction were listed in Appendix Table S1.

## Animals

*Cetn3*$^{flox/+}$ mice (C57BL/6 background) were acquired from GemPharmatech and housed in a specific pathogen-free (SPF) animal facility at Tongji University. *Cetn3*$^{flox/flox}$ mice were generated by mating *Cetn3*$^{flox/+}$ males with *Cetn3*$^{flox/+}$ females, which were then crossed with *Emx1-Cre* mice to produce *Cetn3*$^{flox/+}$ *Emx1-Cre* mice. Subsequently, these *Cetn3*$^{flox/+}$;*Emx1-Cre* mice were crossed with *Cetn3*$^{flox/flox}$ mice to obtain *Cetn3* cKO mice. All the animal experiments mentioned in this study were conducted under the approval of Experimental animal ethics and use Committee of Tongji University.

## Cell culture

Feeder-free H9 and H1 hESC lines (WiCell), and human iPSC line ZSSY001 (Nuwacell) were cultured on Matrigel (Corning)-coated plates using mTeSR medium (Stemcell Technologies, #85850) at 37 °C with 5% $CO_2$, with medium replacement daily. HEK 293T cells were cultured in Dulbecco's Modified Eagle's Medium (DMEM) (Gibco, #C11995500BT) supplemented with 10% fetal bovine serum (FBS), 1×GlutaMax (Invitrogen, #35050-038), and 1% penicillin/streptomycin (P/S) (Sigma, #P0781) at 37 °C with 5% $CO_2$. Transfection was conducted using polyethylenimine L (PEI) (Thermo Fisher, #BMS1003-A), following the manufacturer's instructions.

## Cell lines

H9 and H1 hESCs were utilized to generate *CETN3*-KO cell lines via CRISPR-Cas9 technology. Electroporation of H9 or H1 cells with two sgRNA plasmids targeting intron 1–2 and intron 2–3, along with a homologous recombination donor plasmid, was performed at a 1:1:4 ratio. Following electroporation, clone screening was conducted using puromycin dihydrochloride and GFP. Cell lines that passed genotyping underwent a second electroporation with pCAG-Cre (Addgene, #125574) to remove the exogenous DNA fragment introduced for clone selection. Clones lacking GFP were selected for further genotyping, and Western blot analysis was employed to confirm the deletion of CETN3 post-genotyping. For *CETN3*-mut iPSCs, the patient-derived mutations were introduced using two homologous recombination plasmids. One plasmid contained one of the mutations, along with a puromycin resistance gene and a GFP reporter, while the other plasmid carried the second mutation, a zeocin resistance gene, and a tdTomato reporter. Clone selection was performed based on resistance to puromycin and zeocin, as well as fluorescence from GFP and tdTomato. All other steps followed the same protocol used for generating the *CETN3*-KO cell lines.

## Human cerebral organoids culture

Feeder-free WT and *CETN3*-KO hESCs (H9 and H1), or WT and *CETN3*-mut iPSCs (ZSSY001) were cultured to generate cerebral

organoids using previously established protocols (Lancaster et al, 2013). Briefly, a single-cell suspension was prepared by dissociating ESCs or iPSCs using EDTA and Accutase sequentially, and then 9000 cells were seeded into low-binding 96-well plates (Corning, #3788) treated with anti-adherent, to form embryoid bodies (EB). ROCK inhibitor Y27632 (50 μM; Selleckchem, #S1049) was added at day 0. Neural induction commenced at day 5 by replacing mTeSR with neural induction (NI) medium containing DMEM/F12 (Invitrogen, #11330-032), 1×Glutamax, 1×N2 Supplement (Invitrogen, #17502048), 1×MEM-NEAA (Sigma, #M7145), 1% P/S and 1 μg/mL Heparin (Sigma, #H3149). EBs were fed with fresh NI medium every 2 days for 6 days. At day 11, EBs were embedded into Matrigel (Corning) droplets and cultured in differentiation medium without vitamin A containing a 1:1 mixture of DMEM/F12 and Neurobasal (Invitrogen, #21103049), 0.5×N2 supplement, 1×B27 supplement without vitamin A (Invitrogen), 2.5 ng/mL insulin (Sigma, #I9278-5ML), 1% P/S, 1×Glutamax, 1×MEM-NEAA. The droplets were transferred into differentiation medium with vitamin A on an orbital shaker at day 21. Medium was changed about once a week, and organoid morphology was monitored throughout the culture period.

## 2D neural stem cell differentiation

Feeder-free and *CETN3*-KO H9 hESCs were subjected to 2D neural stem cell differentiation using the Dual-SMAD inhibition method (Shi et al, 2012). In short, cells were dissociated into single cells by EDTA and Accutase, and seeded into Matrigel-coated 24-well plates at a density of 300,000 cells/well in mTeSR medium with 10 μM Y27632. Neural induction was performed the following day if the cells have reached 95–100% confluence by replacing mTeSR medium with 2D neural induction medium containing a 1:1 mixture of DMEM/F12 and Neurobasal, 0.5×N2 supplement, 1×B27 supplement with vitamin A (Invitrogen, #17504044), 2.5 ng/mL insulin, 1% P/S, 1×Glutamax, 1×MEM-NEAA, 1 μM Dorsomorphin (Selleckchem, #S1067), and 10 μM SB431542 (Selleckchem, #S7840). Cells were fed with fresh 2D neural induction medium every day for 12 days to obtain NS/PCs.

## Immunofluorescence

Organoids were fixed in 4% paraformaldehyde overnight at 4 °C and then dehydrated sequentially in 15%, 20%, and 30% sucrose solution. Following dehydration, the tissues were embedded in OCT compound (Sakura) and quickly frozen with dry ice. Cryosectioning of the organoids was performed at a thickness of 16 μm using a Leica CM3050S cryostat. Antigen retrieval was carried out prior to antibody incubation by immersing the samples in an antigen recovery solution (1 mM EDTA, 5 mM Tris at pH 8.0) at ~90 °C for 10 minutes and cooled to room temperature (RT). Subsequently, the sections were blocked in 4% BSA and 0.1% Triton X-100 in PBS for 1 hour at RT, before being incubated with the desired antibodies overnight at 4 °C. Samples were incubated with secondary antibodies for 1.5 hours at RT the following day. Finally, DAPI was used to label the nucleus, and Fluoromount-G (SouthernBiotech, #0100-01) was employed to mount the slides.

For cell slides, cells were seeded in Matrigel-treated 24-well plates with coverslips. The next day, the medium was removed, and 4% paraformaldehyde was added to fix the cells for 20 minutes. After three washes with PBS, the cells were blocked in 4% BSA and 0.1% Triton X-100 in PBS for 1 hour at RT. Subsequently, the cells were incubated with the desired antibodies at a proper dilution rate overnight at 4 °C. Then, the cells were washed three times with PBS following secondary antibody incubation for 1.5 hours at RT, protected from light. DAPI staining was then performed, and the cells were mounted on slides using Fluoromount-G. Antibodies used for immunofluorescence staining were listed in Appendix Table S2.

Observation of the slides was conducted using a Carl Zeiss confocal laser scanning microscope.

## Co-Immunoprecipitation and Western blot

### CoIP
HEK 293T cells were seeded into six-well plates, and transfected with the desired tagged plasmids 24 hours after seeding. After 48 hours, the cells were harvested and lysed in lysis buffer, followed by rotation at 4 °C for 1 hour. The cell lysate was then subjected to sonication for complete lysis. The lysate was subsequently centrifuged at 12,000 × g for 10 minutes at 4 °C. A portion of the supernatant was reserved as the input sample, while the remainder was divided into three equal parts and incubated with Myc-, Flag-, and IgG-magnetic beads for 4 to 5 hours at 4 °C; the beads had been pre-prepared the previous day. Excess antibodies were removed by washing with wash buffer (0.1% Triton X-100, 50 mM Tris-Cl, pH 7.4, 150 mM NaCl, or 500 mM NaCl). Finally, SDS-loading buffer was added to the input and immunoprecipitated samples, followed by boiling at 98 °C for 10 minutes.

### General protein sample preparation
Cells or organoids were harvested and washed with pre-cooled PBS, followed by treatment with 1×SDS-loading buffer to induce protein lysis. The lysates were then boiled at 98 °C for 10 minutes and centrifuged at 12,000×g for 15 minutes at 4 °C. The resulting supernatants were collected as the protein samples.

### Western blot
The protein samples were loaded onto a 10–15% SDS-PAGE gel for electrophoresis. Following separation, the proteins were transferred onto PVDF membranes. The membranes were then blocked with 5% skim milk and incubated with primary antibodies overnight at 4 °C. Subsequently, the membranes were incubated with secondary antibodies for 2 hours at RT. Antibodies used for immunoprecipitation and western blot were listed in Appendix Table S2.

## EdU staining

Click-iT Edu Imaging Kit (Invitrogen, #C10338) was used for Edu imaging following the manufacturer's instructions. Briefly, cells were plated on coverslips coated with Matrigel at the desired density on the first day. The following day, half of the medium was replaced with fresh medium containing 20 μM EdU, and the cells were incubated for a proper time (30 minutes for ESCs, 3 hours for NS/PCs). Afterward, the cells were fixed in 4% paraformaldehyde and permeabilized in 0.5% Triton X-100 in PBS. Then, fresh Click-iT reaction cocktails were prepared and added to each well. The cells were incubated for 30 minutes at RT, shielded from light. Finally, DAPI was utilized to label the nucleus, and Fluoromount-G was applied to mount the slides.

**The paper explained**

**Problem**

Primary microcephaly, a rare congenital condition characterized by reduced brain size, occurs due to impaired neurogenesis during brain development. Through whole-exome sequencing, we identified compound heterozygous loss-of-function mutations in CENTRIN 3 (CETN3) in a 5-year-old patient with primary microcephaly. However, no studies to date have explored the role of CETN3 in neurodevelopment, especially neocortical development.

**Results**

In this study, we generated CETN3-knockout human embryonic stem cell lines and CETN3-mutant induced pluripotent stem cell lines using CRISPR-Cas9 and Cre-LoxP technologies. These cell lines were then differentiated into neural stem/progenitor cells (NS/PCs) via 2D culture and into cerebral organoids through 3D culture. The CETN3-deficient organoids exhibited reduced brain size, recapitulating the phenotype observed in the patient and thereby confirming the pathogenicity of CETN3 loss-of-function. Next, to uncover the molecular basis of CETN3 function, we employed RNA sequencing, immuno-fluorescence, and flow cytometry. These analyses revealed that CETN3 regulates neurogenesis by modulating NS/PC proliferation, differentiation, apoptosis, and RNA splicing through multiple mechanisms: (1) CETN3 deficiency alters the orientation of the mitotic spindles in NS/PCs, biasing them toward differentiation rather than proliferation; (2) CETN3 deficiency impairs the cell cycle progression, thereby reducing NS/PC proliferative capacity; (3) CETN3 deficiency induces centrosome overduplication, leading to increased NS/PC apoptosis; and (4) CETN3 deficiency affects the expression of genes involved in centriole biogenesis, ciliogenesis, and cell cycle regulation by influencing RNA splicing.

**Impact**

In this study, we investigated the role of CETN3 in cerebral cortex development to elucidate its pathogenic potential and the underlying molecular mechanisms. This study provides the first evidence of a critical role for CETN3 in mammalian neurodevelopment and elucidates its underlying molecular mechanisms. We demonstrated that CETN3 regulates the balance between NS/PC proliferation and differentiation by modulating mitotic spindle orientation, cell cycle progression, centrosome duplication, and RNA splicing. These findings underscore the essential role of CETN3 in human neocortical development, advance our understanding of the genetic etiology of primary microcephaly, and offer new perspectives for clinical diagnosis and therapeutic strategies.

## Flow cytometry

Cells in the logarithmic growth phase were incubated with EdU for an appropriate duration at 37 °C. The cells were then dissociated into a single-cell suspension, and cell numbers were equalized across samples. Subsequently, the cells were fixed in 4% paraformaldehyde and permeabilized with 0.5% Triton X-100 in PBS. The cells were then sequentially incubated with Click-iT reaction cocktails and DAPI working solution. Following PBS washes and filtration through a 40 μM filter, the samples were analyzed using flow cytometry instruments.

## RNA-seq and analysis

For organoids, three organoids with good morphology at day 45 were pooled as one sample replicate, and three replicates for each cell line were processed for RNA-seq at the Novogene company. For 2D-differentiated NS/PCs, four replicates were differentiated at the same time and processed for RNA-seq at day 12. Read depth was 6G for each sample. RNA integrity was assessed using the RNA Nano 6000 Assay Kit of the Bioanalyzer 2100 system (Agilent Technologies). RNA sample library preparations were sequenced on an Illumina Novaseq platform, and 150 bp paired-end reads were generated. For data processing, sequencing adapters was cut by fastp to attain clean data first, and data quality was checked by fastqc. Clean data with high quality were aligned to the reference genome hg38 using Hisat2 v2.0.5. FPKM of each gene was calculated by featureCounts v1.5.0-p3. Differential expression analysis of the two groups was processed with DESeq2 v1.20.0, and genes with an adjusted $P$ value <0.05 and log2foldchange >0.5 were assigned as differentially expressed. GO and KEGG enrichment analysis were performed by the clusterProfiler package. GO terms and KEGG pathways with corrected $p$ value <0.05 were considered significantly enriched by differentially expressed genes. Splicing events were identified using rMATS v4.2.0, with events having a false discovery rate (FDR) <0.05 and IncLevelDifference 0.15 were considered statistically significant.

## Statistical analysis

Image data were analyzed using ImageJ software. Specifically, the "Analyze-Measure" tool was used for area measurement, the "Cell Counter" tool for cell counting, and the "Plot Lanes" tool for density analysis of Western blot and RT-PCR. Numerical data were analyzed using GraphPad Prism 8. Statistical significance was determined using one-way ANOVA (three groups comparison) or unpaired $t$-test (two groups comparison), with results presented as mean ± SEM. Significance levels were set at ns (not significant), $*P < 0.05$, $**P < 0.01$, $***P < 0.001$, and $****P < 0.0001$. All experiments were conducted in triplicate.

## Data availability

The raw sequence data reported in this paper have been deposited in the Genome Sequence Archive (Chen et al, 2021) in National Genomics Data Center, China National Center for Bioinformation / Beijing Institute of Genomics, Chinese Academy of Sciences (GSA-Human: HRA008739) that are publicly accessible at https://ngdc.cncb.ac.cn/gsa-human.

The source data of this paper are collected in the following database record: biostudies:S-SCDT-10_1038-S44321-025-00302-7.

## Peer review information

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

## Acknowledgements

This study was supported by grants from the National Key R&D Program of China (2021YFA1100400 and 2020YFA0112500); the National Natural Science Foundation of China (32271019, 32300810, and 12411530079); the China National Postdoctoral Program for Innovative Talents (BX20220230); the Shanghai Sailing Program (22YF1436100); the Natural Science Foundation of Shanghai Municipality (22ZR1462600); the Major Scientific and Technological Projects for Collaborative Prevention and Control of Birth Defects in Hunan Province (2023SK4053); the Fundamental Research Funds for the Central Universities (22120250374); the Peak Disciplines (Type IV) of Institutions of Higher Learning in Shanghai; Work in VK's laboratory is funded by the Human Technopole Early Career Fellowship Program (HT-ECF Programme); work in RX's laboratory is funded by the Royal Society (IEC \NSFC\233033).

## Author contributions

**Jing Xu**: Data curation; Formal analysis; Investigation; Visualization; Methodology; Writing—original draft; Writing—review and editing. **Xiao Mao**: Conceptualization; Resources; Supervision; Writing—review and editing. **Zhen Liu**: Resources; Supervision; Writing—review and editing. **Na Jiang**: Data curation; Formal analysis. **Xin E Wong**: Data curation; Formal analysis. **Deng Liu**: Data curation; Formal analysis. **Yuan Wang**: Formal analysis. **Huaizhe Zhan**: Formal analysis. **Shiyi Liu**: Data curation; Formal analysis. **Jiayao Yu**: Writing—review and editing. **Ruiying Yuan**: Validation; Writing—review and editing. **Qingran Bai**: Writing—review and editing. **Xianshu Bai**: Validation; Writing—review and editing. **Wenhui Huang**: Validation; Writing—review and editing. **Ruoxiao Xie**: Validation; Writing—review and editing. **Veronica Krenn**: Validation; Writing—review and editing. **Frank Kirchhoff**: Validation; Writing—review and editing. **Hua Wang**: Conceptualization; Supervision; Validation; Writing—review and editing. **Zhenming Guo**: Conceptualization; Supervision; Funding acquisition; Validation; Methodology. **Shan Bian**: Conceptualization; Supervision; Funding acquisition; Validation; Investigation; Writing—review and editing.

Source data underlying figure panels in this paper may have individual authorship assigned. Where available, figure panel/source data authorship is listed in the following database record: biostudies:S-SCDT-10_1038-S44321-025-00302-7.

## Disclosure and competing interests statement

This study has been filed a patent application for use of this disease model in future preclinical investigations.

# Expanded View Figures

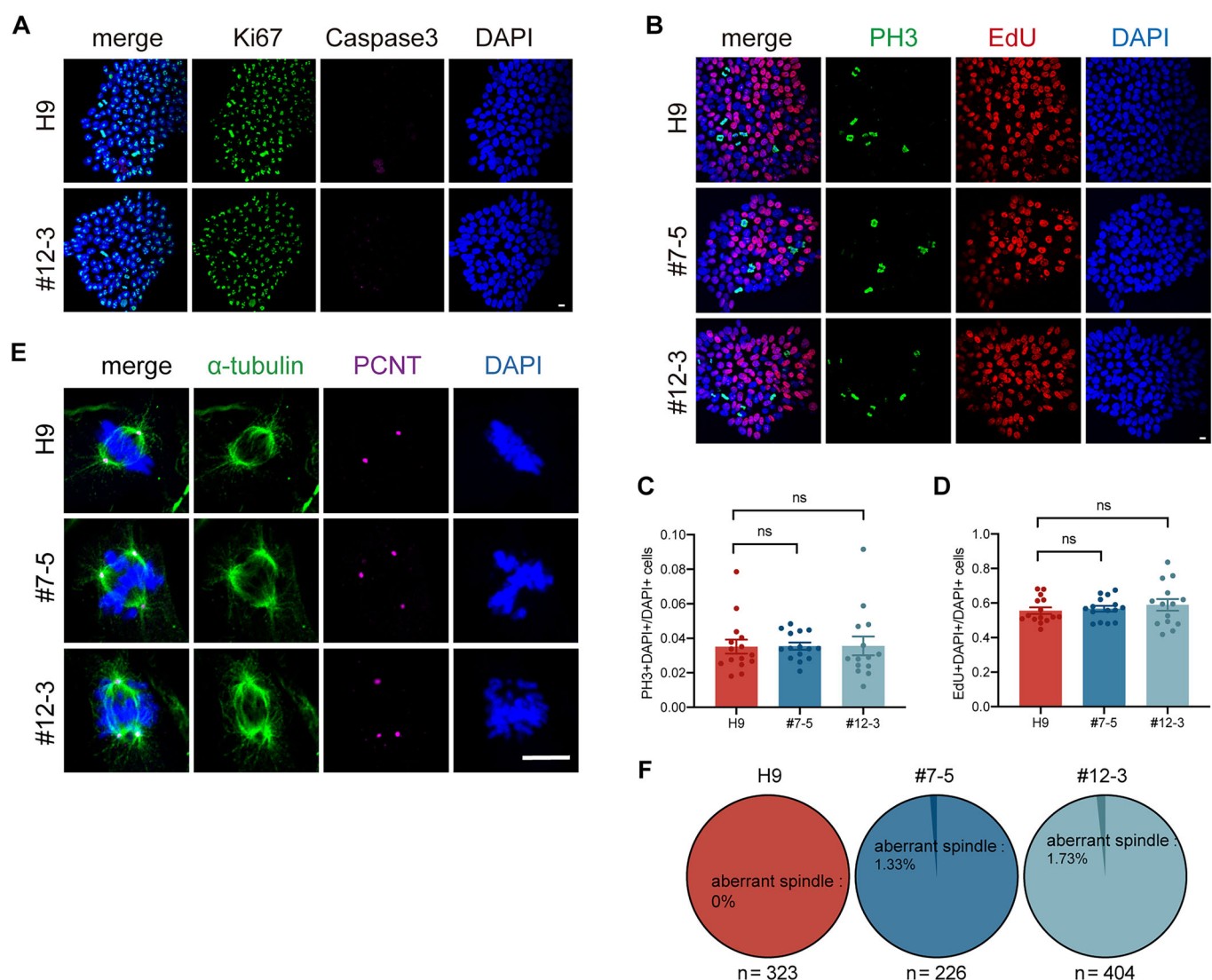

**Figure EV1. Deletion of CETN3 doesn't affect proliferation and apoptosis in ESCs.**

(A, B) Immunofluorescence of ESCs. Markers for proliferation, Ki67, EdU, PH3, and apoptosis, cleaved caspase-3, were stained. ESCs were labeled with EdU for 0.5 h. Scale bar: 10 μm. (C, D) Statistics of PH3 and EdU positive cells in (B). Each dot represented a clone (H9, $n = 15$; #7-5, $n = 15$; #12-3, $n = 14$). The cell during anaphase or telophase was considered a single cell. Data were shown as mean ± SEM. One-way ANOVA was used for differential analysis. ns: not significant. (E) Immunofluorescence of ESCs. Markers for microtubule, α-tubulin, and centrosome, PCNT, were stained. Representative cells undergoing mitosis were magnified for each cell line. Scale bar: 10 μm. (F) Statistics for the proportion of aberrant spindles in each cell line. Red: H9; blue: #7-5; green: #12-3. The darker sections in each pie chart indicated aberrant spindles, with the numbers representing their respective proportions. Data were collected from three independent experiments. Source data are available online for this figure.

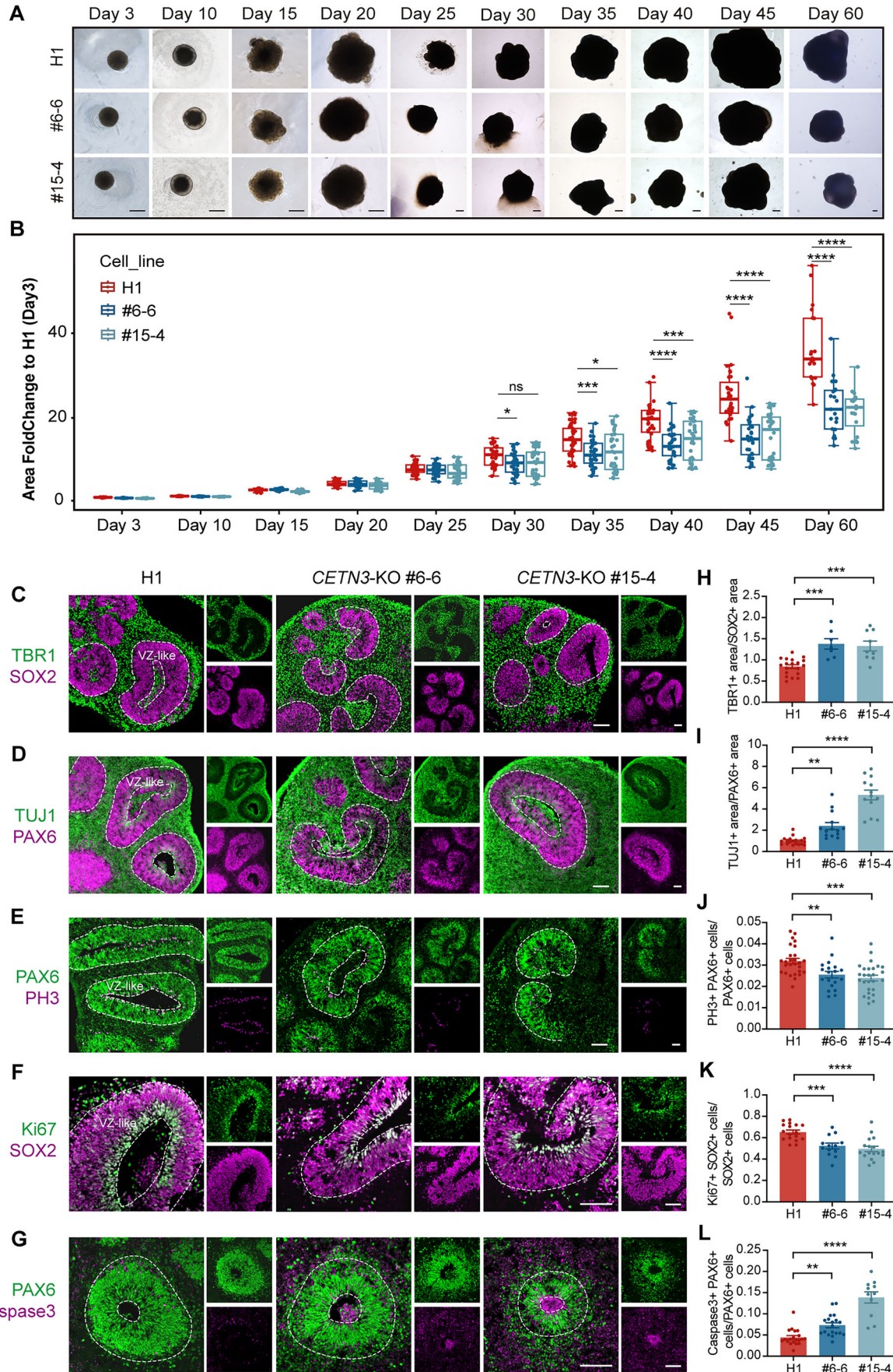

**Figure EV2.  Deficiency of CETN3 interferes with differentiation and proliferation of NS/PCs in hCOs derived from H1.**

(A) Representative images showing the morphology and size of organoids at various stages of culture. Scale bar: 400 μm. (B) Quantification of organoid size at different culture time points. Data were derived from two independent batches and normalized to H1 (day 3). Each plot represents one organoid (day 3: $n = 16$; day 10: $n = 15$; day 15: H1, $n = 34$, #6-6, $n = 32$, #15-4, $n = 37$; day 20: H1, $n = 38$, #6-6, $n = 35$, #15-4, $n = 37$; day 25: H1, $n = 34$, #6-6, $n = 31$, #15-4, $n = 33$; day 30: H1, $n = 34$, #6-6, $n = 33$, #15-4, $n = 34$; day 35: H1, $n = 40$, #6-6, $n = 34$, #15-4, $n = 38$; day 40: H1, $n = 32$, #6-6, $n = 33$, #15-4, $n = 36$; day 45: H1, $n = 31$, #6-6, $n = 28$, #15-4, $n = 32$; day 60: H1, $n = 21$, #6-6, $n = 21$, #15-4, $n = 20$). Boxes represent the IQR from the first to third quartile, with the median shown as a horizontal line. Whiskers extend to the most extreme data points within 1.5× IQR; outliers beyond this range are shown as individual dots. Day 30: $P = 0.011$ (H1 vs. #6-6), $P = 0.059$ (H1 vs. #15-4); day 35: $P = 0.000164$ (H1 vs. #6-6), $P = 0.012$ (H1 vs. #15-4); day 40: $P < 0.0001$ (H1 vs. #6-6), $P = 0.000295$ (H1 vs. #15-4); day 45: $P < 0.0001$ (H1 vs. #6-6), $P < 0.0001$ (H1 vs. #15-4); day 60: $P < 0.0001$ (H1 vs. #6-6), $P < 0.0001$ (H1 vs. #15-4). Statistical analysis was conducted using one-way ANOVA. *$P < 0.05$; ***$P < 0.001$; ****$P < 0.0001$; ns not significant. (C, D) Immunofluorescence staining of cryosections from day 45 hCOs. NS/PCs were labeled with SOX2 or PAX6, while neurons were identified using TBR1 or TUJ1. Scale bar: 100 μm. VZ: ventricular zone. (E, F) Immunofluorescence staining of cryosections from day 45 hCOs. Markers used included PAX6 or SOX2 for NS/PCs, PH3 and Ki67 for cell proliferation. Scale bar: 100 μm. (G) Immunofluorescence staining of cryosections from day 45 hCOs. Markers used included PAX6 for NS/PCs, cleaved caspase-3 for cell apoptosis. Scale bar: 100 μm. (H–L) Quantification of cell number or staining area for each marker. Each dot represented an individual rosette or several adjacent rosettes (H: H1, $n = 19$, #6-6, $n = 7$, #15-4, $n = 11$; I: H1, $n = 19$, #6-6, $n = 14$, #15-4, $n = 13$; J: H1, $n = 27$, #6-6, $n = 19$, #15-4, $n = 27$; K: H1, $n = 17$, #6-6, $n = 13$, #15-4, $n = 17$; L: H1, $n = 20$, #6-6, $n = 19$, #15-4, $n = 11$). Data were collected from organoids across two independent experiments, with results presented as mean ± SEM. (H) $P = 0.0002$ (H1 vs. #6-6), $P = 0.0002$ (H1 vs. #15-4); (I) $P = 0.0021$ (H1 vs. #6-6), $P < 0.0001$ (H1 vs. #15-4); (J) $P = 0.0051$ (H1 vs. #6-6), $P = 0.0001$ (H1 vs. #15-4); (K) $P = 0.0009$ (H1 vs. #6-6), $P < 0.0001$ (H1 vs. #15-4); (L) $P = 0.0048$ (H1 vs. #6-6), $P < 0.0001$ (H1 vs. #15-4). Differential analysis was performed using one-way ANOVA. **$P < 0.01$; ***$P < 0.001$; ****$P < 0.0001$. Source data are available online for this figure.

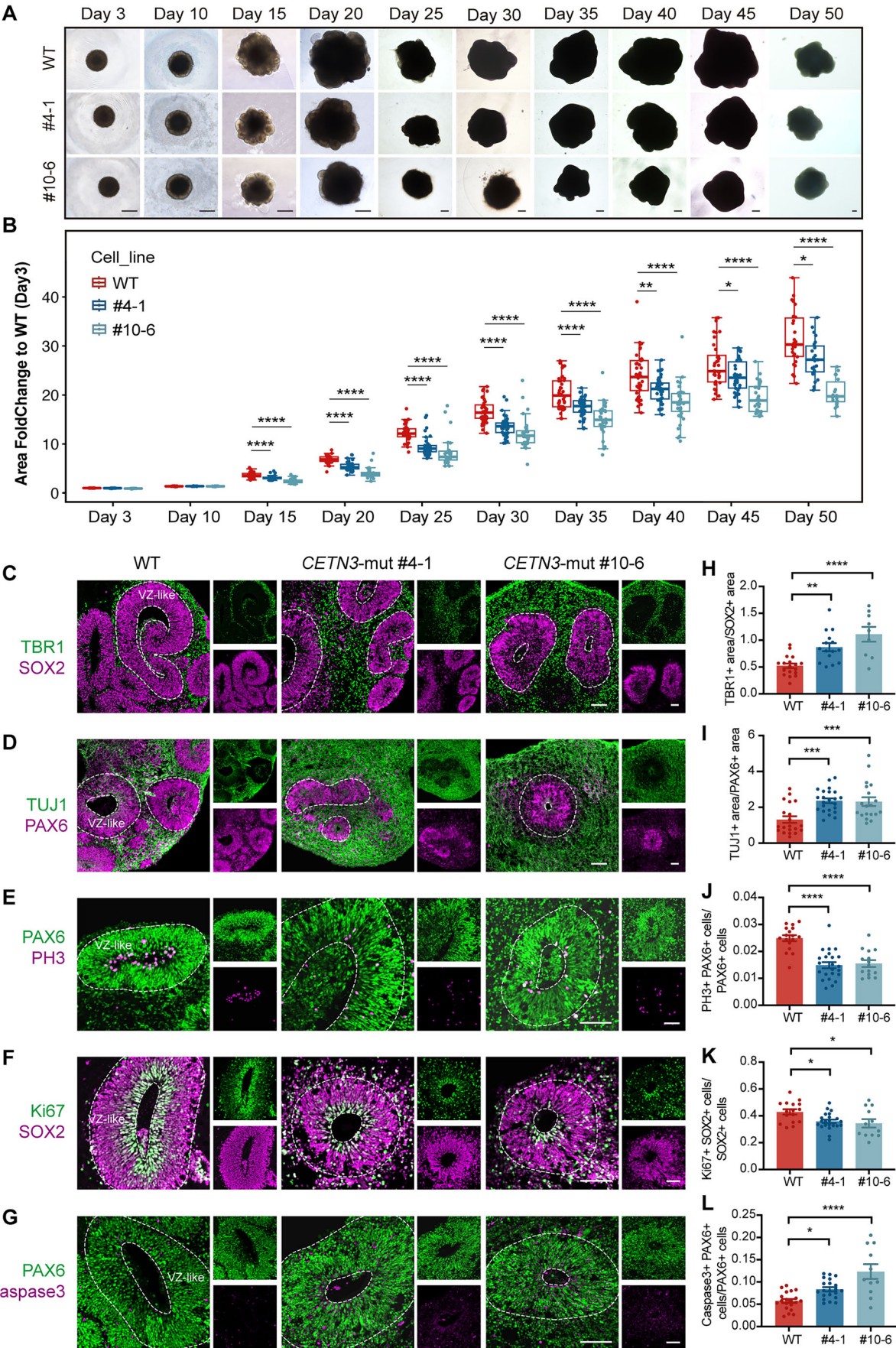

◀ **Figure EV3.  Loss-of-function mutations in *CETN3* impair NS/PC differentiation and proliferation in iPSC-derived hCOs.**

(A) Representative images showing the morphology and size of organoids at various stages of culture. Scale bar: 400 μm. (B) Quantification of organoid size at different culture time points. Data were derived from two independent batches and normalized to WT group (day 3). Each plot represents one organoid (day 3: $n = 16$; day 10: $n = 16$; day 15: WT, $n = 39$, #4-1, $n = 39$, #10-6, $n = 37$; day 20: WT, $n = 35$, #4-1, $n = 41$, #10-6, $n = 25$; day 25: WT, $n = 37$, #4-1, $n = 41$, #10-6, $n = 29$; day 30: WT, $n = 40$, #4-1, $n = 41$, #10-6, $n = 32$; day 35: WT, $n = 40$, #4-1, $n = 40$, #10-6, $n = 35$; day 40: WT, $n = 40$, #4-1, $n = 42$, #10-6, $n = 36$; day 45: WT, $n = 36$, #4-1, $n = 34$, #10-6, $n = 32$; day 50: WT, $n = 26$, #4-1, $n = 23$, #10-6, $n = 18$). Boxes represent the IQR from the first to third quartile, with the median shown as a horizontal line. Whiskers extend to the most extreme data points within 1.5× IQR; outliers beyond this range are shown as individual dots. Day 15: $P < 0.0001$ (WT vs. #4-1), $P < 0.0001$ (WT vs. #10-6); day 20: $P < 0.0001$ (WT vs. #4-1), $P < 0.0001$ (WT vs. #10-6); day 25: $P < 0.0001$ (WT vs. #4-1), $P < 0.0001$ (WT vs. #10-6); day 30: $P < 0.0001$ (WT vs. #4-1), $P < 0.0001$ (WT vs. #10-6); day 35: $P < 0.0001$ (WT vs. #4-1), $P < 0.0001$ (WT vs. #10-6); day 40: $P = 0.002$ (WT vs. #4-1), $P < 0.0001$ (WT vs. #10-6); day 45: $P = 0.032$ (WT vs. #4-1), $P < 0.0001$ (WT vs. #10-6); day 50: $P = 0.012$ (WT vs. #4-1), $P < 0.0001$ (WT vs. #10-6). Statistical analysis was conducted using one-way ANOVA. *$P < 0.05$; **$P < 0.01$; ****$P < 0.0001$; ns not significant. (C, D) Immunofluorescence staining of cryosections from day 45 hCOs. NS/PCs were labeled with SOX2 or PAX6, while neurons were identified using TBR1 or TUJ1. Scale bar: 100 μm. VZ ventricular zone. (E, F) Immunofluorescence staining of cryosections from day 45 hCOs. Markers used included PAX6 or SOX2 for NS/PCs, PH3 and Ki67 for cell proliferation. Scale bar: 100 μm. (G) Immunofluorescence staining of cryosections from day 45 hCOs. Markers used included PAX6 for NS/PCs, cleaved caspase-3 for cell apoptosis. Scale bar: 100 μm. (H–L) Quantification of cell number or staining area for each marker. Each dot represented an individual rosette or several adjacent rosettes (H: WT, $n = 18$, #4-1, $n = 15$, #10-6, $n = 9$; I: WT, $n = 18$, #4-1, $n = 21$, #10-6, $n = 17$; J: WT, $n = 17$, #4-1, $n = 23$, #10-6, $n = 15$; K: WT, $n = 16$, #4-1, $n = 22$, #10-6, $n = 12$; L: WT, $n = 20$, #4-1, $n = 21$, #10-6, $n = 11$). Data were collected from organoids across two independent experiments, with results presented as mean ± SEM. (H) $P = 0.0026$ (WT vs. #4-1), $P < 0.0001$ (WT vs. #10-6); (I) $P = 0.0003$ (WT vs. #4-1), $P = 0.0009$ (WT vs. #10-6); (J) $P < 0.0001$ (WT vs. #4-1), $P < 0.0001$ (WT vs. #10-6); (K) $P = 0.0199$ (WT vs. #4-1), $P = 0.0153$ (WT vs. #10-6); (L) $P = 0.0172$ (WT vs. #4-1), $P < 0.0001$ (WT vs. #10-6). Differential analysis was performed using one-way ANOVA. *$P < 0.05$; **$P < 0.01$; ***$P < 0.001$; ****$P < 0.0001$. Source data are available online for this figure.

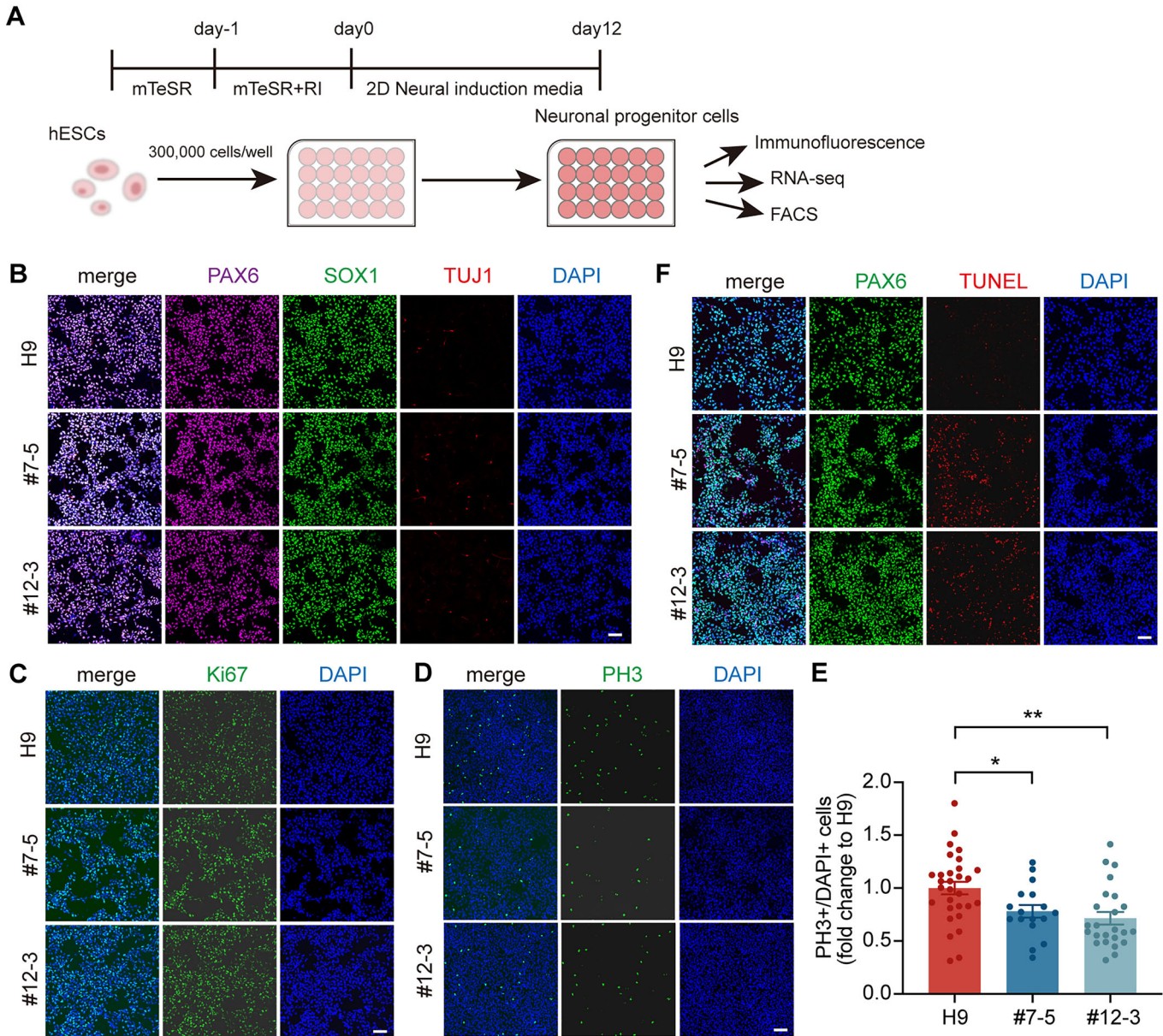

**Figure EV4. Deletion of CETN3 affects proliferation and apoptosis in NS/PCs.**

(A) Technological process for 2D differentiation. On day −1, 300,000 cells were plated in a Matrigel-coated 24-well plate, and the ROCK inhibitor was added. On day 0, the media were switched from mTesR to 2D neural induction media. Fresh 2D neural induction media were replaced daily until day 12 to obtain NS/PCs. The NS/PCs were then used for immunofluorescence, RNA-seq, and flow cytometry analysis. (B) Immunofluorescence of NS/PCs. Markers for NS/PCs, PAX6 and SOX1, and neurons, TUJ1, were stained. Scale bar: 100 μm. (C, D) Immunofluorescence of NS/PCs. Markers for proliferation, Ki67 and PH3, were stained. Scale bar: 100 μm. (E) Statistics of PH3-positive cells in (D). Each dot indicated the statistical result of an individual image (H9, $n = 30$; #7-5, $n = 17$; #12-3, $n = 24$). The cell during anaphase or telophase was considered a single cell. Data were collected from three independent experiments, with results presented as mean ± SEM. $P = 0.0347$ (H9 vs. #7-5), $P = 0.0017$ (H9 vs. #12-3). One-way ANOVA was used for differential analysis. $*P < 0.05$; $**P < 0.01$. (F) TUNEL technique to observe apoptosis in NS/PCs, PAX6 was co-stained. Scale bar: 100 μm. Source data are available online for this figure.

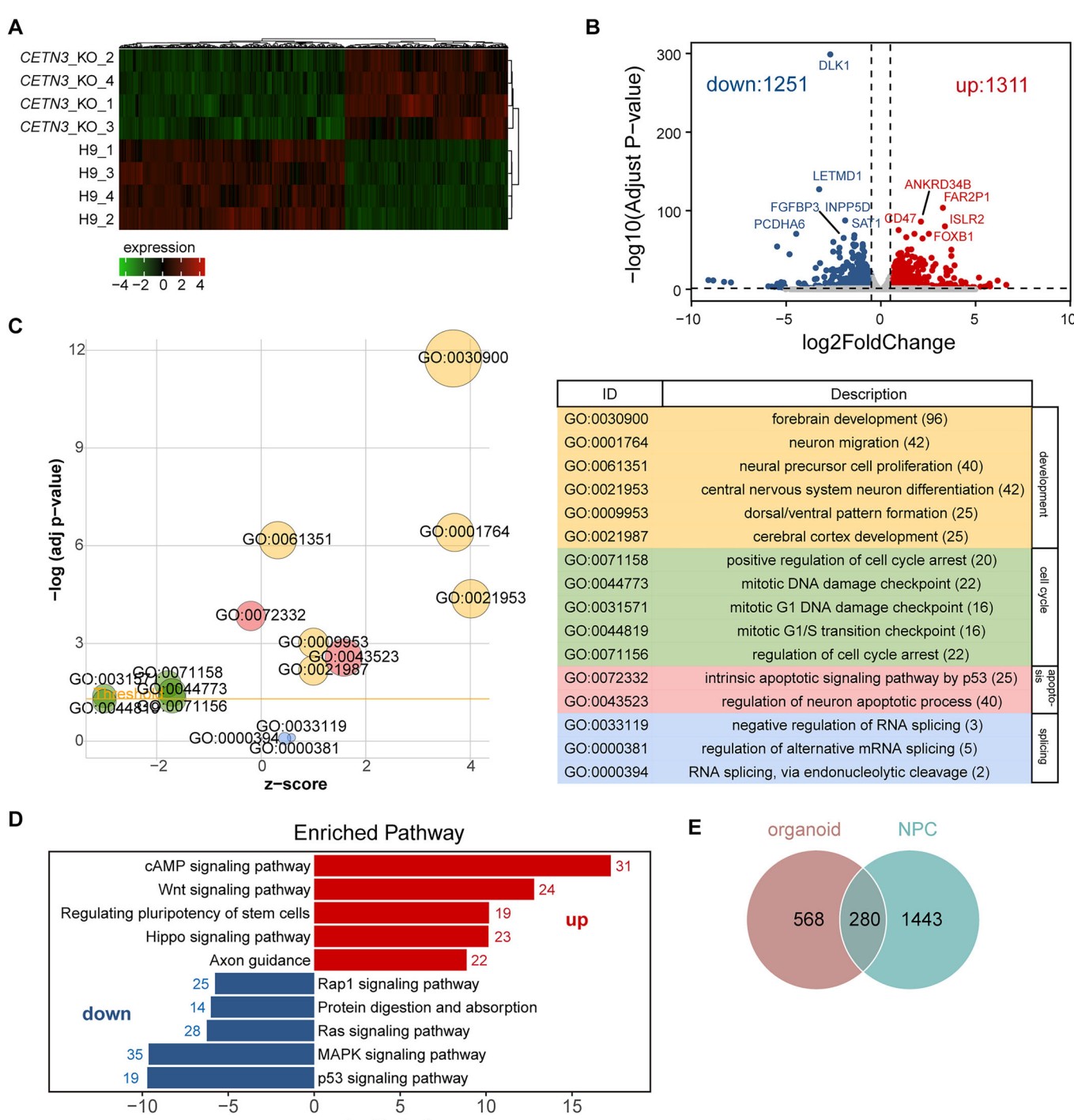

**Figure EV5. Deletion of CETN3 disrupts the expression of genes related to neuronal proliferation, differentiation, apoptosis, and splicing in NS/PCs.**

(A) Heatmap showed DEGs (*padj* <0.05, |log2FC| >0.5) between H9 and *CETN3*-KO NS/PCs at day 12 from 2D differentiation. Four replicates for each group. (B) Volcano plot showed DEGs between H9 and *CETN3*-KO NS/PCs. The top five genes were labeled in the plot. Statistical significance of differential gene expression was determined in DESeq2 using a Wald test, with *p* values adjusted for multiple comparisons by the Benjamini–Hochberg procedure. (C) GO_MF enrichment bubble plot (left), and description for the GO terms (right). The GO terms were manually classified into four categories according to their biological functions, and gene counts for each term were labeled behind. (D) KEGG enrichment results. Pathways related to brain development were selected. The numbers beside each bar were gene counts for each pathway. Red, upregulated pathway; blue, downregulated pathway. Enrichment analysis was performed using clusterProfiler to identify GO terms and KEGG pathways enriched among differentially expressed genes. Gene set *P* values were calculated using a hypergeometric test and adjusted for multiple comparisons using the Benjamini–Hochberg method. (E) Intersection of differentially spliced genes in organoids (day 45) and NS/PCs from 2D differentiation.

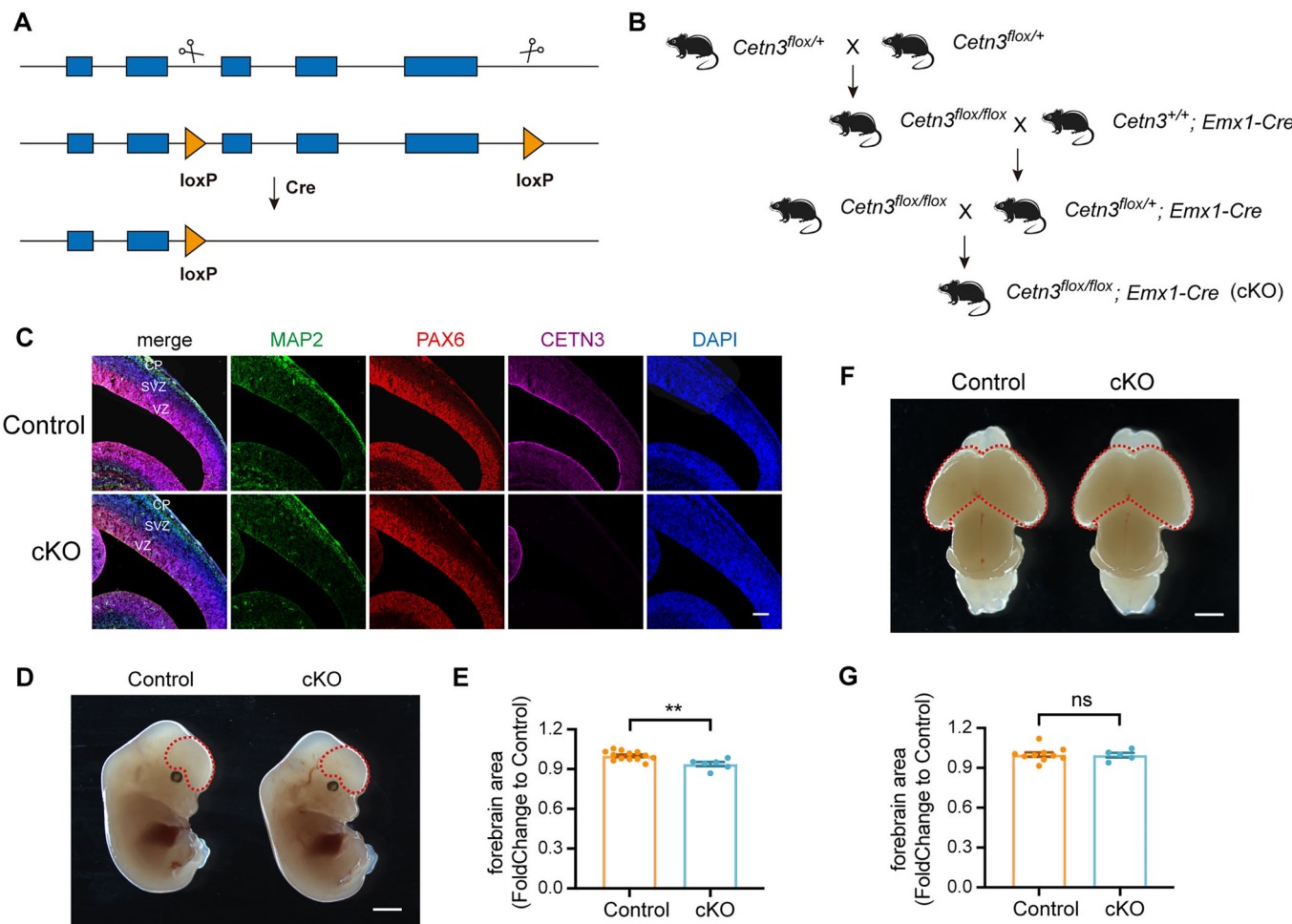

**Figure EV6.   Conditional deletion of *Cetn3* in the mouse forebrain affects brain development exclusively during the embryonic stage.**

(A) Schematic of the *Cetn3* knockout strategy in mice. Two loxP elements were inserted into intron2-3 and 3′ UTR, respectively, enabling Cre-mediated excision of exons 3–5. (B) Breeding strategy used to generate *Cetn3*-cKO mice. (C) Immunofluorescence staining of coronal cryosections from E13.5 mouse brains. Markers for neurons (MAP2), NS/PCs (PAX6), and CETN3 were co-stained. VZ ventricular zone, SVZ subventricular zone, CP cortical plate. Scale bar: 100 μm. (D) Representative images of E13.5 embryos from control and *Cetn3*-cKO mice. Forebrains were outlined with red dashed lines. Scale bar: 1 mm. (E) Quantification of forebrain area in E13.5 mice. Data were normalized to control. Control, $n = 13$; cKO, $n = 6$ (from three litters). Data shown as mean ± SEM. $P = 0.0025$ (Control vs. cKO). An unpaired *t*-test was used for differential analysis. **$P < 0.01$. (F) Representative images of P0 brains from control and *Cetn3*-cKO mice. Forebrains were outlined with red dashed lines. Scale bar: 1 mm. (G) Quantification of forebrain area in P0 mice. Data were normalized to control. Control, $n = 10$; cKO, $n = 5$ (from three litters). Data were shown as mean ± SEM. $P = 0.8759$ (Control vs. cKO). An unpaired *t*-test was used for differential analysis. ns not significant. Source data are available online for this figure.

