## [Peer Review File · EMBO Molecular Medicine]

CETN3 Deficiency Induces Microcephaly by Disrupting NPC Cell Fate through Impaired Centrosome Assembly and RNA Splicing

Jing Xu, Xiao Mao, Zhen Liu, Na Jiang, Xin E Wong, Deng Liu, Yuan Wang, Huaizhe Zhan, Shiyi Liu, Jiayao Yu, Ruiying Yuan, Qingran Bai, Xianshu Bai, Wenhui Huang, Ruoxiao Xie, Veronica Krenn, , Frank Kirchhoff, Qing-Ran Bai, Hua Wang, Zhenming Guo, and Shan Bian

Corresponding authors: Shan Bian (shan_bian@tongji.edu.cn) , zhenming guo (21310192@tongji.edu.cn), Hua Wang (FX20240002@csu.edu.cn)

Review Timeline:

Submission Date:	28th Oct 24
Editorial Decision:	13th Dec 24
Revision Received:	2nd Jun 25
Editorial Decision:	22nd Jul 25
Revision Received:	11th Aug 25
Accepted:	18th Aug 25

Editor: Jingyi Hou

Transaction Report:

13th Dec 2024

Dear Shan,

Thank you for submitting your work to EMBO Molecular Medicine. I would like to apologise for the slow process, which was due to the late arrival of reviewers' reports. We have now heard back from the three reviewers who agreed to evaluate your manuscript. You will see from the comments below that the Reviewers find the manuscript to be of interest. They raise, however, several important points, which should be convincingly addressed in a revision of this work.

I think that the recommendations of the reviewers are rather clear so there is no need to repeat the points listed below. In particular, both Reviewers #1 and #2 had some concerns about the current model system and suggested that the study would benefit from including an additional cell line, along with some specific recommendations. During our pre-decision cross-commenting process (in which the reviewers are given the chance to make additional comments, including on each other's reports), Reviewer #2 added, "Knocking out Centrin in another ES line will most likely reveal the same phenotype. In this case, I will prefer an iPSC line. As I mentioned, the authors must have used a patient line. At the least, they should express that the patient mutation is an iPS cell line and see if it disrupts the cellular functions observed in ES KO." This concern should be addressed carefully.

As you may already know, our editorial policy allows in principle a single round of major revision, so it is essential to provide responses to the reviewers' comments that are as complete as possible. Please feel free to contact me in case you would like to discuss in further detail any of the issues raised by the reviewers.

EMBO Molecular Medicine has a "scooping protection" policy, whereby similar findings that are published by others during review or revision are not a criterion for rejection. Should you decide to submit a revised version, I do ask that you get in touch after six months if you have not completed it, to update us on the status.

Please also contact us as soon as possible if similar work is published elsewhere. If other work is published, we may not be able to extend the revision period beyond six months.

I look forward to receiving your revised manuscript.

Kind regards,
Jingyi

Jingyi Hou
Senior Editor
EMBO Molecular Medicine

We require:

2) Individual production quality figure files as .eps, .tif, .jpg (one file per figure). For guidance, download the 'Figure Guide PDF': (<https://www.embopress.org/page/journal/17574684/authorguide#figureformat>).

3) A .docx formatted letter INCLUDING the reviewers' reports and your detailed point-by-point responses to their comments. As part of the EMBO Press transparent editorial process, the point-by-point response is part of the Review Process File (RPF), which will be published alongside your paper.

4) A complete author checklist, which you can download from our author guidelines (<https://www.embopress.org/page/journal/17574684/authorguide#submissionofrevisions>). Please insert information in the checklist that is also reflected in the manuscript. The completed author checklist will also be part of the RPF.

6) It is mandatory to include a 'Data Availability' section after the Materials and Methods. Before submitting your revision, primary datasets produced in this study need to be deposited in an appropriate public database, and the accession numbers and database listed under 'Data Availability'. Please remember to provide a reviewer password if the datasets are not yet public (see <https://www.embopress.org/page/journal/17574684/authorguide#dataavailability>).

.

12) Author contributions: You will be asked to provide CRediT (Contributor Role Taxonomy) terms in the submission system. These replace a narrative author contribution section in the manuscript.

13) A Conflict of Interest statement should be provided in the main text.

14) Every published paper now includes a 'Synopsis' to further enhance discoverability. Synopses are displayed on the journal webpage and are freely accessible to all readers. They include a short stand first (maximum of 300 characters, including space) as well as 2-5 one-sentence bullet points that summarize the paper. Please write the bullet points to summarize the key NEW findings. They should be designed to be complementary to the abstract - i.e. not repeat the same text. We encourage inclusion of key acronyms and quantitative information (maximum of 30 words / bullet point). Please use the passive voice. Please attach these in a separate file or send them by email, we will incorporate them accordingly.

15) All Materials and Methods need to be described in the main text using our 'Structured Methods' format. According to this format, the Methods section includes a Reagents and Tools Table (listing key reagents, experimental models, software and relevant equipment and including their sources and relevant identifiers) followed by a Methods and Protocols section describing the methods, ideally using a step-by-step protocol format. The aim is to facilitate adoption of the methodologies across labs.

Please download and fill our Reagents and Tools Table template (.docx), which you can find in our author guidelines: <https://www.embopress.org/page/journal/17574684/authorguide#structuredmethods>

***** Reviewer's comments *****

Referee #1 (Comments on Novelty/Model System for Author):

Authors used cerebral organoid model from one cell line to recapitulate CETN3 loss-of-function variants identified from a single patient. It will be more convincing if authors could re-phenocopy the microcephalic features using cerebral organoid model derived from another cell line (for instance, another CETN3-KO cell line derived from other human embryonic stem cell lines or induced pluripotent stem cell lines) or patient iPSCs with only CETN3 mutations.

Referee #1 (Remarks for Author):

In this manuscript, Xu and colleagues explored the roles of CETN3, a centrosome gene, in neurodevelopment, and identified it as a microcephaly gene. Using human embryonic stem cell-derived cerebral organoid model, authors recapitulated the microcephalic phenotype observed in a 5-year-old patient carrying biallelic loss-of-function variants in CETN3, and investigated the potential underlying cellular mechanisms. They observed that CETN3-deletion in cerebral organoids led to impaired proliferation, increased cell apoptosis of NPCs, and enhanced pre-mature neuronal differentiation. In addition, authors uncovered a role of CETN3 in RNA splicing. There are several major and minor concerns that potentially improved the significance of this study.

Major points:

1. Authors used cerebral organoid model from one cell line to recapitulate CETN3 loss-of-function variants identified from a single patient. It will be more convincing if authors could re-phenocopy the microcephalic features using cerebral organoid model derived from another cell line (for instance, another CETN3-KO cell line derived from other human embryonic stem cell lines or induced pluripotent stem cell lines) or patient iPSCs with only CETN3 mutations.
2. Since CETN3 plays important roles in general centriole biogenesis, it is possible that loss of cent3 might affect the pluripotency of hESCs. In the current study, authors tested the pluripotency by immunostaining of pluripotent markers SOX2 and OCT4. However, pluripotency could also be affected by the expression level of these markers. Please verify the pluripotency upon the deletion of CETN3.
3. Figure 1G-H, the expression of CETN3 in human is much higher than in mouse. Does that indicate the dosage-related function is more important in humans?
4. The description of Figure 2A is likely not the same as showing in figure. It looks like the system is based on Crispr/Cas9 but not Cre-dependent deletion. It's very confusing.
5. In Figure 3D, how did authors define these genes' categories? Is that based on the GO terms? Plus, Figure 3C and D, how

many genes were included in each term or pathway?

6. Figure 4C and I, the localization of TBR2+ progenitors were not very classic and the IPCs should be in SVZ. Why did TBR2 increase in KO hCOs?

7. Line 262, the authors claimed "Additionally, increased apoptosis may be attributed to abnormal centrosome duplication and defective cell division". How about the ratio of Caspase 3 staining in Fig 4L? Are they related?

8. Why DEGs in Fig EV3 is much more than in Fig3B?

9. Figure 5, authors suggest that CENT3 regulates the cell cycle via interacting with USP44. To further prove the hypothesis, authors should conduct western blot analysis testing the downstream cell cycle-related genes potentially regulated by USP44.

Minor points:

1. Figure 1A,B and Appendix Figure S1A,B are not clear presented. Authors should replace them with higher resolution or bigger font.

2. Figure 1I and J, please show zoom images to indicate the clear centrosome localization of CENT3.

3. Line 108, authors should clarify the "normal length". What length is it?

4. Figure 5I, no color legends.

Referee #2 (Remarks for Author):

Shan Bian's work identifies a biallelic mutation in Centrin3 in a microcephaly patient. The authors then eliminated the gene in ES cells and modeled the defective cellular functions that could underlie the patient's microcephaly syndrome. The work is straightforward and very clean. The article reads well. The images are of high quality, and the controls are appropriately checked.

Conceptually, I do not see anything shocking. The findings are just expected and supported by the standard experiments of high quality. Centrin 3 is a centrosome protein and one of the best-characterized molecules conserved throughout evolution. Plus, it is also highly expressed in the human brain. Eliminating it causing the cell cycle defect, polarity defect etc.. is all expected.

It would make sense if the authors had analyzed the iPS cells derived from the patient. Eliminating centrin 3 in ES cells does not offer any unexpected /surprising findings.

The authors can overexpress one of those mutations in WT ES cells. Do they observe a dominant mutant behavior causing microcephaly phenotypes?

The authors have shown very nice phenotypes, but the molecular mechanisms (as it is a centrosome protein) are lacking. As it is a centrosomal protein, one would expect to see if there are structural changes, whether there is centrosome fragmentation (it appears the case in 2D NPCs), and whether there is a connection to defective primary cilia. Addressing these questions would make this work attractive for cell biologists focusing on organelle mechanisms. However, this could be too much to ask as my questions are core to the molecular mechanisms. While this can be discussed between the authors and the editor, getting deeper into such molecular mechanisms will make this work very attractive. At the least, the authors must discuss centrosomes and primary cilia as microcephaly is primarily a centrosome disease.

Even without these demanded experiments, I agree that this work deserves EMBO Mol Med.

Some of my minor comments are below.

The title is misleading. It reads as if CETN3 deficiency is required for centrosome assembly and splicing.

Like this, there are many funny sentences. At the end of the introduction, it reads....

"Our findings suggest that CETN3 regulates brain size, either directly by impairing centrosome formation required for NPC proliferation, differentiation, and apoptosis, or indirectly, through interaction with RNA splicing machinery responsible for processing mRNAs of centriole-associated genes?."

Does it mean that Centrin regulates brain size by impairing centrosome formation? And apoptosis?

Where does the antibody bind? It would help explain it schematically.

Figure 4, bar graphs. How did the authors calculate the area for each marker-specified region?

Referee #3 (Comments on Novelty/Model System for Author):

The manuscript by Xu J et al is in my opinion an important study as it identifies the first human CENTRIN3 mutation in congenital microcephaly and demonstrates the effects of loss of this protein on neurogenesis using a hES cell-derived brain organoid model. As such, it should be highly cited.

However, I find it somewhat surprising and unusual that a clinical case is described without any clinical image of the patient or at least of her brain (MRI), MRI which has obviously been carried out since the authors give precise details in the text.

Experiments and organoid models to demonstrate the causality of CENTRIN3 loss on the fate of neural progenitors (and thus on the microcephaly phenotype) are convincing and of high quality. The identification of RNA splicing anomalies brings real mechanistic originality to the study, but I'm not sure that their involvement in the phenotype is fully demonstrated in the experiments presented.

In my opinion, this study deserves to be published in EMBO Mol Med if the authors can address the points raised. I thank you again for giving me the opportunity to expertise a work for EMBO Mol Med, and I hope my input will help in the editorial decision.

Referee #3 (Remarks for Author):

In this manuscript, Jing Xu and colleagues identify compound heterozygous loss-of-function mutations in CETN3, the gene encoding CENTRIN 3 in a child with primary microcephaly and demonstrate, by developing brain organoids in which CETN3 has been invalidated by CRISPR-Cas9, that loss of CENTRIN 3 affects the neural fate of progenitors and in particular their proliferation and neuronal differentiation. As is often the case in primary microcephaly, cell cycle and mitotic spindle abnormalities are identified in progenitors, leading to increased apoptosis or premature differentiation, and thus the author conclude to a direct influence of CENTRIN 3 on NPC cell fate. Using an RNA-seq approach, they also identify numerous mRNA splicing abnormalities, including in genes associated with centriolar regulation, and present arguments in favour of an indirect influence of CENTRIN 3 on the cell cycle.

This manuscript is very well written, the experiments are of high quality and report the first human mutations associated with CENTRIN 3 while providing a satisfactory mechanistic explanation for the microcephalic phenotype. The occurrence of splicing abnormalities is also very interesting, even if the demonstration of their direct involvement in the phenotype made in the manuscript is less convincing. I believe this paper is certainly worthy of publication if the authors can improve it by taking into account the following issues.

1. Given that this is the first clinical description associated with CETN3 and that there is only one patient, it is of the utmost importance to provide the MRI of the patient's brain that the authors describe but do not show. This will be of invaluable help in identifying new patients in the future and refining the phenotype.
2. The authors use the Lancaster protocol to generate brain organoids, an unguided protocol that generates a large number of brain areas in the same organoid with a well-described high variability. To conclude that neuronal differentiation is premature in CETN3-KO organoids, the authors use the neuronal marker TBR1 and the intermediate progenitor marker TBR2, and quantify TBR1/PAX6 and TBR2/PAX6 ratios. However, the unguided protocol can generate highly variable quantities of PAX6-positive SVZ that are negative for TBR1 and TBR2 and this might not be due to the absence of CENTRIN 3. Premature differentiation should be assessed with a neuronal marker that avoids this problem, e.g. by quantifying neurons with the HuCD marker. It would also be more convincing to show the increase in differentiation at an earlier stage than 45 days, when few neurons are still expected, for example at day 30.
3. The authors identify RNA splicing abnormalities in NPCs, including in cell cycle-associated genes, but provide no evidence that these abnormalities directly disrupt the cell cycle in their model and are therefore responsible for the fate defects observed. I understand that this is difficult to demonstrate, but the authors should at least add a paragraph to the discussion highlighting the existence of very interesting previous studies linking splicing system anomalies to the occurrence of microcephaly and ciliary anomalies (for example PMID 30093551 and PMID37612280 but there are several others).
4. A CETN3 mouse model was published in 2015 by G Ying (and cited by the authors in the introduction) which mentions a ciliogenesis-related phenotype in the CETN3 and CETN2 double KO but no cilium phenotype in the CETN3 single KO. The authors should comment on this difference with their results in the discussion and raise the link of CETN3 with CETN2 in centriolar regulation.
5. The Co-IP experiment with USP44 adds nothing to the story, as this interaction provides no input on the consequences of CETN3 loss-of-function on the function of this protein, which is moreover not discussed again afterwards. In my opinion, Figure 5G should be removed from the manuscript.
6. The introduction (line 59) mentions 8 loci associated with hereditary primary microcephaly, but it's much more than that.

Please correct this by referring to Farcy's recent paper (PMID 37443841), which contains an updated table.

7. Did exome sequencing reveal only the variants identified in CETN3 and nothing at all in any other gene? If other variants predicted to be pathogenic have been identified, please mention them and explain how they can be excluded. This information is lacking in the methods section.

8. The incidence of microcephaly indicated in the first sentence of the discussion corresponds rather to the prevalence of genetic microcephalies, please correct. The prevalence of all-cause microcephaly is much higher (estimated at between 0.15% and 1.53%, depending on the study, the region of the world and the definition of microcephaly (differences in Standard Deviation Cut-off), see PMID27623840))

9. In the first sentence of the abstract, change "microcephaly" to "primary microcephaly" because you're talking specifically about congenital microcephaly

**** Reviewer's comments ****

Referee #1 (Comments on Novelty/Model System for Author):

Authors used cerebral organoid model from one cell line to recapitulate CETN3 loss-of-function variants identified from a single patient. It will be more convincing if authors could re-phenocopy the microcephalic features using cerebral organoid model derived from another cell line (for instance, another CETN3-KO cell line derived from other human embryonic stem cell lines or induced pluripotent stem cell lines) or patient iPSCs with only CETN3 mutations.

Referee #1 (Remarks for Author):

In this manuscript, Xu and colleagues explored the roles of CETN3, a centrosome gene, in neurodevelopment, and identified it as a microcephaly gene. Using human embryonic stem cell-derived cerebral organoid model, authors recapitulated the microcephalic phenotype observed in a 5-year-old patient carrying biallelic loss-of-function variants in CETN3, and investigated the potential underlying cellular mechanisms. They observed that CETN3-deletion in cerebral organoids led to impaired proliferation, increased cell apoptosis of NS/PCs, and enhanced pre-mature neuronal differentiation. In addition, authors uncovered a role of CETN3 in RNA splicing. There are several major and minor concerns that potential improved the significance of this study.

Major points:

1. Authors used cerebral organoid model from one cell line to recapitulate CETN3 loss-of-function variants identified from a single patient. It will be more convincing if authors could re-phenocopy the microcephalic features using cerebral organoid model derived from another cell line (for instance, another CETN3-KO cell line derived from other human embryonic stem cell lines or induced pluripotent stem cell lines) or patient iPSCs with only CETN3 mutations.

Answer: Thank you for your valuable suggestion. In response, we have generated two additional cell lines: CETN3-KO H1 hESCs and a patient-specific CETN3-mutant iPSC line derived from a wild-type iPSC line. CETN3 was knocked out in H1 hESCs using the same strategy as previously employed for generating CETN3-KO lines in H9 cells. Additionally, we introduced the biallelic patient-derived mutations in CETN3 into wild-type iPSCs to establish the CETN3-mut iPSC line. Both successfully generated cell lines were used to culture human cerebral organoids (hCOs), which recapitulated the microcephalic phenotypes. These results are presented in the newly added Appendix Figure S4, Figure EV2, and Figure EV3.

2. Since CENT3 play important roles in general centriole biogenesis, it is possible that loss of cent3 might the pluripotency of hESCs. In the current study, authors tested the pluripotency by immunostaining of pluripotent markers SOX2 and OCT4. However,

pluripotency could also be affected by the expression level of these markers. Please verify the pluripotency upon the deletion of CETN3.

Answer: Thank you for your valuable suggestion. In response, we performed immunostaining for NANOG in the CETN3-KO ES cells, as shown in the revised Appendix Figure S2. Additionally, we assessed the pluripotency of the cells by qPCR, as demonstrated in revised Figure 2C, analyzing the expression of key pluripotency-related genes, including *NANOG*, *OCT4*, *SOX2*, *DNMT3B*, *TERT*, and *REX1*. The results indicate that the loss of CETN3 does not affect the pluripotency of hESCs.

3. Figure 1G-H, the expression of CETN3 in human is much higher than in mouse. Does that indicate the dosage-related function is more important in humans?

Answer: We would like to thank you for this importance question. In response to the concern regarding the stronger CETN3 signal in human samples compared to mouse samples, we acknowledge that differences in microscope settings, antibody affinity, or gene expression levels could potentially influence the observed results. To address this, we re-acquired the images using identical imaging parameters. The results revealed that the human sample maintained a slightly stronger CETN3 signal (see the Response Figure 1). Upon closer examination, a higher density of CETN3 was observed at the apical surface, suggesting an increased abundance of centrosomes. This observation aligns with the higher number of neural stem/progenitor cells (NS/PCs) in the human ventricular zone (VZ). While we cannot entirely rule out species-specific differences in antibody affinity, the data supports the hypothesis that the slightly stronger CETN3 signal in human samples reflects an increased centrosome abundance, consistent with the higher NS/PC density in the human VZ.

Response Figure 1. The immunostaining of CETN3 antibody on human and mouse brain tissues.

4. The description of Figure 2A is likely not the same as showing in figure. It looks like the system is based on Crispr/Cas9 but not Cre-dependent deletion. It's very confusing.

Answer: We apologize for any confusion in our previous description. To clarify, our experimental approach utilized a two-step genome-editing process to generate *CETN3*-deletion hESC lines. First, we used Crispr/Cas9 and homology recombination to replace the Exon2 of *CETN3* with a loxP sites-flanked CAG promoter-transcript selection element (loxP-CAG-eGFP-IRES-Puro-loxP). Second, we employed the Cre/loxP system to remove the exogenous DNA fragment introduced for clone selection (see revised Figure 2A). We have updated the relevant descriptions in both the main text and the Methods section accordingly, and modified the Figure 2A.

5. In Figure 3D, how did authors define these genes' categories? Is that based on the GO terms? Plus, Figure 3C and D, how many genes were included in each term or pathway?

Answer: We are sorry for the unclear description in our initial submission. The gene categories in Figure 3D were determined using GO terms, our expertise, and existing literature. Detailed information about certain genes is included in the main text. For well-known neurogenesis-related genes, such as *PAX6*, *MAP2*, and *DCX*, detailed explanations were omitted. We also added gene counts for each term or pathway to the revised Figure 3 and Figure EV5.

6. Figure 4C and I, the localization of TBR2+ progenitors were not very classic and the IPCs should be in SVZ. Why did TBR2 increase in KO hCOs?

Answer: Thank you for pointing this out. Typically, the majority of TBR2+ intermediate progenitor cells (IPCs) are localized in the subventricular zone (SVZ), with a smaller subset present in the ventricular zone (VZ). In our experiment, their distribution appears accurately. In Figure 4C (new Figure 4D), the white dashed line delineates the VZ, and most TBR2+ IPCs are observed at the outer boundary of the VZ, corresponding to the SVZ. We hypothesize that the increased number of TBR2+ IPCs results from an elevated rate of asymmetric divisions in radial glial cells (RGCs), which leads to a pre-mature neuronal differentiation, in *CETN3*-KO hCOs, as shown in Figures 5A and 5B.

7. Line 262, the authors claimed "Additionally, increased apoptosis may be attributed to abnormal centrosome duplication and defective cell division". How about the ratio of Caspase 3 staining in Fig 4L? Are they related?

Answer: Thank you for your question. Caspase 3 staining in Fig. 4L (new Fig. 4G) demonstrates an increased proportion of apoptotic NS/PCs in *CETN3*-KO hCOs, as evidenced by a higher number of Caspase 3-positive cells in the KO group. Similarly, the TUNEL assay conducted on 2D-differentiated NS/PCs confirms elevated apoptosis in the *CETN3*-KO group (Fig. EV4F). However, the underlying cause of this increased apoptosis remains unclear. Given the presence of aberrant spindle structures observed in 2D-differentiated NS/PCs, we hypothesize that abnormal centrosome duplication may contribute to the observed apoptosis.

8. Why DEGs in Fig EV3 is much more than in Fig3B?

Answer: Thank you for your concerns. We applied the same threshold ($|\log_2FC| > 0.5$, $p_{adj} < 0.05$) to identify DEGs in both hCOs and 2D-differentiated NS/PCs. The greater number of DEGs in NS/PCs compared to hCOs indicates that CETN3 loss has a more pronounced effect on NS/PCs. As a centrosome-associated protein, CETN3 is more active in proliferative NS/PCs than in non-proliferative neurons. In hCOs, which contain both cell types, DEGs specific to NS/PCs may be masked by neuronal contributions in bulk RNA-seq data.

9. Figure 5. authors suggest that CENT3 regulates the cell cycle via interacting with USP44. To further prove the hypothesis, authors should conduct western blot analysis testing the downstream cell cycle-related genes potentially regulated by USP44.

Answer: Thanks for your suggestion. Since USP44 could regulate the spindle assembly checkpoint by regulating the activity of Cdc20-anaphase promoting complex/cyclosome (Cdc20-APC/C) E3 ubiquitin ligase (PMID: 17443180, PMID: 21853124), we have tested Cdc20-APC/C activity by assessing the levels of CyclinB1, a known substrate of Cdc20-APC/C, via western blot analysis. The results have been inserted into revised Figure 5H-J.

Minor points:

1. Figure 1A,B and Appendix Figure S1A,B are not clear presented. Authors should replace them with higher resolution or bigger font.

Answer: We apologize for the low quality of the original images. In the revised Figure 1 and Appendix Figure S1, we have enhanced the image resolution and increased the font size for improved readability.

2. Figure 1I and J, please show zoom images to indicate the clear centrosome localization of CENT3.

Answer: We believe you may be referring to Figures 1J and 1K, as Figure 1I does not contain co-localization information. The zoomed-in images for Figures 1J and 1K were already provided in the lower panels of the original figures. However, we apologize for missing this detail in the figure legend. We have now included this information in the revised manuscript.

3. Line 108, authors should clarify the "normal length". What length is it?

Answer: We apologize for the vague description. By 'normal,' we refer to the average values expected for age, gender, and population. This has been clarified in the revised manuscript.

4. Figure 5I, no color legends.

Answer: We apologize for missing the color information in Figure 5I (new Figure 5L); this information has now been added to the revised figure legend, also for Figure 5B and Figure EV1F.

Referee #2 (Remarks for Author):

Shan Bian's work identifies a biallelic mutation in Centrin3 in a microcephaly patient. The authors then eliminated the gene in ES cells and modeled the defective cellular functions that could underlie the patient's microcephaly syndrome. The work is straightforward and very clean. The article reads well. The images are of high quality, and the controls are appropriately checked.

Conceptually, I do not see anything shocking. The findings are just expected and supported by the standard experiments of high quality. Centrin 3 is a centrosome protein and one of the best-characterized molecules conserved throughout evolution. Plus, it is also highly expressed in the human brain. Eliminating it causing the cell cycle defect, polarity defect etc.. is all expected.

It would make sense if the authors had analyzed the iPS cells derived from the patient. Eliminating centrin 3 in ES cells does not offer any unexpected /surprising findings.

Answer: Thank you for your essential suggestion. We fully agree that using patient-derived iPSCs will largely improve our study. However, we apologize for our inability to obtain iPSCs from the patient since patient's family declined our request. Instead, we have generated two additional cell lines, including *CETN3*-KO H1 hESCs and genetically engineered iPSC line carrying patient-specific *CETN3*-mutants. Specifically, *CETN3* was knocked out in H1 hESCs using the same strategy employed for generating *CETN3*-KO lines in H9 cells. In addition, the biallelic patient-carrying mutations in *CETN3* were introduced into wild-type iPSCs to establish *CETN3*-mut iPSCs. These successfully generated cell lines were used to culture human cerebral organoids (hCOs), which recapitulated the microcephalic phenotypes. The results are presented in the newly added Appendix Figure S4, Figure EV2, and Figure EV3.

The authors can overexpress one of those mutations in WT ES cells. Do they observe a dominant mutant behavior causing microcephaly phenotypes?

Answer: Thank you for the suggestion. However, we believe overexpressing the mutations in WT ES cells would not have an effect, since the patient's mutations are two deletion mutations: one with 2-bp deletion and the other with 4-bp deletion, both causing frameshift and leading to premature STOP codon. If these mutations were overexpressed in WT ES cells, the resulting mutant proteins would most likely be degraded, preventing the detection of any dominant-negative effects. Therefore, we consider that introducing the mutations into the endogenous genome of WT ESCs or iPSCs would be a more appropriate approach. Our results showed that the hCOs derived from the iPSCs carrying patient-specific mutations reproduced the microcephalic symptoms of the patient.

The authors have shown very nice phenotypes, but the molecular mechanisms (as it is a centrosome protein) are lacking. As it is a centrosomal protein, one would expect to see if there are structural changes, whether there is centrosome fragmentation (it appears the case in 2D NS/PCs), and whether there is a connection to defective primary cilia. Addressing these questions would make this work attractive for cell biologists focusing on organelle mechanisms. However, this could be too much to ask as my questions are core to the molecular mechanisms. While this can be discussed between the authors and the editor, getting deeper into such molecular mechanisms will make this work very attractive. At the least, the authors must discuss centrosomes and primary cilia as microcephaly is primarily a centrosome disease.

Answer: Thank you very much for your suggestion. We agree that it will make the study more complete if we can show the structural defects of centrosome or primary cilia using electron microscope. We have tried three times to analyze the structure of the centrosome and cilia in both NS/PCs and organoids using transmission electron microscope but failed to obtain clear images of these structures. Although we are not able to provide the structural results of these organelles in the revised manuscript, we have showed that the density of primary cilia in both CETN3-KO cerebral organoids (see Appendix Figure S5 and the Response Figure 2A) and embryonic mouse brains deficient for *Cetn3* (see the Response Figure 2B) is significantly reduced.

Response Figure 2. The primary cilia density of NSCs from CETN3-KO hCOs (**A**) and *Cetn3*-cKO mouse brains (**B**) is significantly reduced compared to the cilia density of their corresponding controls.

Even without these demanded experiments, I agree that this work deserves EMBO Mol Med.

Some of my minor comments are below.

The title is misleading. It reads as if CETN3 deficiency is required for centrosome assembly and splicing.

Answer: We apologize for the confusion. The title has been revised to “CETN3 Deficiency Induces Microcephaly by Disrupting NS/PC Cell Fate through Impaired Centrosome Assembly and RNA Splicing”

Like this, there are many funny sentences. At the end of the introduction, it reads....

"Our findings suggest that CETN3 regulates brain size, either directly by impairing centrosome formation required for NS/PC proliferation, differentiation, and apoptosis, or indirectly, through interaction with RNA splicing machinery responsible for processing mRNAs of centriole-associated genes?.

Does it mean that Centrin regulates brain size by impairing centrosome formation? And apoptosis?

Answer: We apologize for the confusion. We have revised the relevant description in the manuscript.

Where does the antibody bind? It would help explain it schematically.

Answer: Apologies, but the manufacturer of the CETN3 antibody does not provide information on its binding site. We have also contacted the manufacturer's scientific support team, but they informed us that the immunogen sequence is proprietary and cannot be disclosed. However, they indicated that the binding site is likely located in the central region of human CETN3. We have added this information in the Figure Legend.

Figure 4, bar graphs. How did the authors calculate the area for each marker-specified region?

Answer: Thank you for your suggestion to clarify our measurement in our study. We calculated the area of the desired regions on organoid slices using Image J. First, we set the scale for each image, then applied the freehand selection tool to outline regions positive for markers such as PAX6. Finally, the “Analyze-Measure” tool was used to quantify the area of the selected regions. We apologize for missing this information in the “Methods” section. It has now been included in the “Statistical Analysis” part of the revised manuscript.

Referee #3 (Comments on Novelty/Model System for Author):

The manuscript by Xu J et al is in my opinion an important study as it identifies the first human CENTRIN3 mutation in congenital microcephaly and demonstrates the effects of

loss of this protein on neurogenesis using a hES cell-derived brain organoid model. As such, it should be highly cited.

However, I find it somewhat surprising and unusual that a clinical case is described without any clinical image of the patient or at least of her brain (MRI), MRI which has obviously been carried out since the authors give precise details in the text.

Experiments and organoid models to demonstrate the causality of CENTRIN3 loss on the fate of neural progenitors (and thus on the microcephaly phenotype) are convincing and of high quality. The identification of RNA splicing anomalies brings real mechanistic originality to the study, but I'm not sure that their involvement in the phenotype is fully demonstrated in the experiments presented.

In my opinion, this study deserves to be published in EMBO Mol Med if the authors can address the points raised. I thank you again for giving me the opportunity to expertise a work for EMBO Mol Med, and I hope my input will help in the editorial decision.

Referee #3 (Remarks for Author):

In this manuscript, Jing Xu and colleagues identify compound heterozygous loss-of-function mutations in CETN3, the gene encoding CENTRIN 3 in a child with primary microcephaly and demonstrate, by developing brain organoids in which CETN3 has been invalidated by CRISPR-Cas9, that loss of CENTRIN 3 affects the neural fate of progenitors and in particular their proliferation and neuronal differentiation. As is often the case in primary microcephaly, cell cycle and mitotic spindle abnormalities are identified in progenitors, leading to increased apoptosis or premature differentiation, and thus the author conclude to a direct influence of CENTRIN 3 on NS/PC cell fate. Using an RNA-seq approach, they also identify numerous mRNA splicing abnormalities, including in genes associated with centriolar regulation, and present arguments in favour of an indirect influence of CENTRIN 3 on the cell cycle.

This manuscript is very well written, the experiments are of high quality and report the first human mutations associated with CENTRIN 3 while providing a satisfactory mechanistic explanation for the microcephalic phenotype. The occurrence of splicing abnormalities is also very interesting, even if the demonstration of their direct involvement in the phenotype made in the manuscript is less convincing. I believe this paper is certainly worthy of publication if the authors can improve it by taking into account the following issues.

1. Given that this is the first clinical description associated with CETN3 and that there is only one patient, it is of the utmost importance to provide the MRI of the patient's brain that the authors describe but do not show. This will be of invaluable help in identifying new patients in the future and refining the phenotype.

Answer: Thanks for your suggestion. We are sorry for not providing this important clinical information in our initial submission. The MRI images have been added to Appendix Figure S1C.

2. The authors use the Lancaster protocol to generate brain organoids, an unguided protocol that generates a large number of brain areas in the same organoid with a well-described high variability. To conclude that neuronal differentiation is premature in CETN3-KO organoids, the authors use the neuronal marker TBR1 and the intermediate progenitor marker TBR2, and quantify TBR1/PAX6 and TBR2/PAX6 ratios. However, the unguided protocol can generate highly variable quantities of PAX6-positive SVZ that are negative for TBR1 and TBR2 and this might not be due to the absence of CENTRIN 3. Premature differentiation should be assessed with a neuronal marker that avoids this problem, e.g. by quantifying neurons with the HuCD marker. It would also be more convincing to show the increase in differentiation at an earlier stage than 45 days, when few neurons are still expected, for example at day 30.

Answer: Thank you for the suggestion. We have performed IF staining for HuC/D in hCOs at both day 35 and day 45. Additionally, PH3 and TBR1 staining was carried out on day 35 hCOs. These results are now included in the revised Figure 4 and Appendix Figure S3.

3. The authors identify RNA splicing abnormalities in NS/PCs, including in cell cycle-associated genes, but provide no evidence that these abnormalities directly disrupt the cell cycle in their model and are therefore responsible for the fate defects observed. I understand that this is difficult to demonstrate, but the authors should at least add a paragraph to the discussion highlighting the existence of very interesting previous studies linking splicing system anomalies to the occurrence of microcephaly and ciliary anomalies (for example PMID 30093551 and PMID37612280 but there are several others).

Answer: Thank you for the suggestion. We have incorporated a description linking RNA splicing to microcephaly in the "Discussion" section and have cited the relevant literature. Additionally, we demonstrated that the loss of CETN3 significantly impaired the deubiquitinase activity of USP49 on H2B in 2D-differentiated NS/PCs, as shown by Western blot analysis (revised Fig. 6L-N). Since USP49 regulates co-transcriptional pre-mRNA splicing through the deubiquitination of histone H2B, these results suggest that CETN3 modulates RNA splicing via its interaction with USP49.

4. A CETN3 mouse model was published in 2015 by G Ying (and cited by the authors in the introduction) which mentions a ciliogenesis-related phenotype in the CETN3 and CETN2 double KO but no cilium phenotype in the CETN3 single KO. The authors should comment on this difference with their results in the discussion and raise the link of CETN3 with CETN2 in centriolar regulation.

Answer: Thank you for this suggestion. We have supplemented this information in our revised "Discussion" section. We think the species difference may account for the observed difference in phenotypes between mice and humans. While human CETN3

and CETN2 share similar functions, they are not redundant and cannot fully compensate for each other. However, the deletion of CETN3 could be compensated by CETN2 in mice. In fact, we also generated forebrain-specific *Cetn3*-ablated mice, which did not exhibit a microcephaly phenotype at birth but show subtle phenotype at E13.5 with data as follows (see Response Figure 3). These results have been added into the “Discussion” and new Figure EV6.

Response Figure 3. *Cetn3*-cKO mouse brains revealed a subtle reduction in size compared to control brains at early developmental stage (A, E13.5), but showed no significant difference in size at birth (B, P0).

5. The Co-IP experiment with USP44 adds nothing to the story, as this interaction provides no input on the consequences of CETN3 loss-of-function on the function of this protein, which is moreover not discussed again afterwards. In my opinion, Figure 5G should be removed from the manuscript.

Answer: Thanks for this suggestion. Since USP44 could regulate the spindle assembly checkpoint by regulating the activity of Cdc20-anaphase promoting complex/cyclosome (Cdc20-APC/C) E3 ubiquitin ligase (PMID: 17443180, PMID: 21853124), we have tested Cdc20-APC/C activity by assessing the levels of CyclinB1, a known substrate of Cdc20-APC/C, via western blot analysis. The results have been inserted into revised figure 5H-J. But we could also remove this result from the revised manuscript according to your suggestion.

6. The introduction (line 59) mentions 8 loci associated with hereditary primary microcephaly, but it's much more than that. Please correct this by referring to Farcy's recent paper (PMID 37443841), which contains an updated table.

Answer: Thank you for the suggestion and correction. We have revised the gene number in the “Introduction” section.

7. Did exome sequencing reveal only the variants identified in CETN3 and nothing at all

in any other gene? If other variants predicted to be pathogenic have been identified, please mention them and explain how they can be excluded. This information is lacking in the methods section.

Answer: We apologize for missing this information in the original manuscript. The details regarding Genetic Variant Analysis have been added in the revised "Methods". Besides, other variants predicted to be pathogenic have been listed in "Appendix table S3".

8. The incidence of microcephaly indicated in the first sentence of the discussion corresponds rather to the prevalence of genetic microcephalies, please correct. The prevalence of all-cause microcephaly is much higher (estimated at between 0.15% and 1.53%, depending on the study, the region of the world and the definition of microcephaly (differences in Standard Deviation Cut-off), see PMID27623840))

Answer: Thank you for the suggestion. We have reviewed the paper you mentioned (PMID: 27623840). The original wording in the paper states: "The prevalence of microcephaly in Europe was 1.53 (95% confidence interval 1.16 to 1.96) per 10,000 births," which corresponds to a prevalence of 0.0153%, rather than 1.53%. We have revised the prevalence of microcephaly in "Discussion" section after referring additional articles (PMID: 34112604, 33229416, 32926077).

9. In the first sentence of the abstract, change "microcephaly" to "primary microcephaly" because you're talking specifically about congenital microcephaly

Answer: Thank you for your suggestion. We have revised the abstract in accordance with your advice.

22nd Jul 2025

Dear Shan,

Thank you for submitting the revised version of your manuscript to EMBO Molecular Medicine. We have now received the reports from the three referees who re-evaluated your work. As you will see below, they are generally satisfied with the revisions. However, before we can proceed with acceptance, we kindly ask you to address the following remaining points:

1. The remaining minor issues mentioned by Referee #3.

On a more editorial level:

2. Please remove the "Author Contributions" section from the manuscript file.

3. Appendix:

- All appendix figures and tables should be compiled into a single PDF file labeled "Appendix".
- Include a table of contents on the first page of the Appendix, with corresponding page numbers.
- Place figure legends directly beneath each figure within the Appendix, and remove these legends from the main manuscript file.
- Provide brief descriptions for each Appendix table.
- Ensure there are callouts in the main text for all Appendix tables.

4. Please provide a 'Synopsis' to further enhance discoverability. Synopses are displayed on the journal webpage and are freely accessible to all readers. They include a short stand first (maximum of 300 characters, including space) as well as 2-5 one-sentences bullet points that summarizes the paper. Please write the bullet points to summarize the key NEW findings. They should be designed to be complementary to the abstract - i.e. not repeat the same text. We encourage inclusion of key acronyms and quantitative information (maximum of 30 words / bullet point). Please use the passive voice. Please attach these in a separate file, we will incorporate them accordingly.

Please also provide a visual abstract to illustrate your article as a PNG file 550 px wide x 300-600 px high.

5. Funding Information: Please ensure that the funding information entered in the journal submission system is consistent with the information provided in the manuscript file.

- There is a discrepancy in the project number for the Fundamental Research Funds for the Central Universities: it appears as 22120250374 in the manuscript, but as 22120240435 in our system. Please correct this inconsistency.
- The Human Technopole Early Career Fellowship Programme (HT-ECF Programme) and the Royal Society (IEC\NSFC\233033) are listed in the manuscript but are missing from the submission system. Please solve this inconsistency.

6. Source Data: Please compress the source data for the Appendix figures and EV figures into a single ZIP folder.

7. "Data Accessibility" Section:

- Please rename this section to "Data Availability."
- Note that this section should only refer to primary datasets; therefore, please remove any descriptions related to previously published data.
- The RNA-seq data generated in this study must be deposited in an appropriate public repository. Please include the name of the database and the corresponding accession numbers within this section.

8. Reference format: Please limit the number of author names listed to 10 followed by et al.

9. Please add a "Disclosure statement and competing interests".

10. Since the study involves human participants: in Methods (and in Author checklist), include a statement that informed consent was obtained from all subjects and that the experiments conformed to the principles set out in the WMA Declaration of Helsinki and the Department of Health and Human Services Belmont Report.

11. Please address the following issues related to figure legends:

- Please note that the exact p values are not provided in the legends of figures 2F, 4H-N; 5D-F, I, J; 6I, J, M, N; EV2 B, H-L; EV3B, H-L; EV4 E, EV6 E; S3D-F; S5A, B, D.
- Please indicate the statistical test used for data analysis in the legends of figures 3B, C, E; 6B-F; EV5 B, D; EV6 E, G
- Please note that the box plots need to be defined in terms of minima, maxima, bounds of box and whiskers, and percentile in the legends of figures 2F, EV2 B; EV3 B
- Please note that information related to n is missing in the legends of figures 6B-H; EV1 C, D; EV2 B, H-L; EV3 B, H-L; EV5 B

- Please note that the error bars are not defined in the legends of figures EV6 E, G.

12. During a standard image analysis, we detected potential aberrations in the figure set, and we would like to clarify these issues before proceeding further with your manuscript. We kindly invite you to check the composition of:

Figure 2E Day 45 - 1st and second cells. The background in both slides appears to be identical (also in the provided source data) but only in these two cells.

Also, please note the change in contrast from day 3 to day 35, showing a white background. For days 45 & 60, a black background. This should remain consistent throughout the figure.

Figure EV6D: cKO image appears to have another image inserted into the background information of a separate image.

Figure EV6F: Contains distinct, clear white circles in the background information - this is consistent with the use of the erasure or cloning stamp tool in Photoshop.

See attached images with Photoshop enhancements activated to highlight the issues shown.

Please provide us with all related source data and a point-by-point response to the issues found. If you make changes to the figure set, please include this in the point-by-point describing what you have changed and why.

Image source data should be provided as one file per figure that contains the original, uncropped and unprocessed scans of all or key gels/microscopy images used in the figure. The file(s) should be labelled with the appropriate figure/panel number, and should display molecular weight markers; further annotation may be useful but is not essential. Source data files will be published online with the article as supplementary "Source Data."

13. Please add the heading "Figure legends" and "Expanded View Figure Legends" to the manuscript text.

14. Move "The paper explained" to the manuscript file.

Thank you for submitting this paper to EMBO Molecular Medicine.

Sincerely,
Jingyi

Jingyi Hou
Senior Editor
EMBO Molecular Medicine

*** Instructions to submit your revised manuscript ***

***** Reviewer's comments *****

Referee #1 (Remarks for Author):

The authors have addressed all my previous comments.

Referee #2 (Remarks for Author):

My comments have not been fully addressed. However, the authors have given suitable explanation and addressed most of the comments raised by other two reviewers. I congratulate the authors for writing such a nice and useful manuscript. I have no reservation to accept the paper for publication.

Referee #3 (Comments on Novelty/Model System for Author):

The new version of the manuscript is solid and well written, the experiments are of high quality, report the first human mutations associated with CENTRIN 3 and provide a coherent and convincing mechanistic explanation for the microcephalic phenotype. The occurrence of splicing abnormalities is also of high interest, and has been properly discussed.

Referee #3 (Remarks for Author):

Thank you for correcting the number of MCPH loci initially estimated at 8 in the first version of the manuscript. However, it is not 52 as indicated in the new version, but rather 32 as indicated in the cited reference (Farcy et al 2023). Please correct again. The other points raised have, in my opinion, been properly addressed.

1. The remaining minor issues mentioned by Referee #3.

Referee #3 (Remarks for Author):

Thank you for correcting the number of MCPH loci initially estimated at 8 in the first version of the manuscript. However, it is not 52 as indicated in the new version, but rather 32 as indicated in the cited reference (Farcy et al 2023). Please correct again. The other points raised have, in my opinion, been properly addressed.

The number of MCPH genes has been corrected to 32 in the introduction.

18th Aug 2025

Dear Shan,

Congratulations on an excellent manuscript, I am pleased to inform you that your manuscript has been accepted for publication in the EMBO Molecular Medicine. Thank you for your comprehensive response to referee concerns. It has been a pleasure to work with you to get this to the acceptance stage.

Sincerely,
Jingyi

Jingyi Hou
Senior Editor
EMBO Molecular Medicine
